# A Prestressed Concrete Cylinder Pipe Broken Wire Detection Algorithm Based on Improved YOLOv5

**DOI:** 10.3390/s25030977

**Published:** 2025-02-06

**Authors:** Haoze Li, Ruizhen Gao, Fang Sun, Yv Wang, Baolong Ma

**Affiliations:** 1College of Mechanical and Equipment Engineering, Hebei University of Engineering, Handan 056038, China; haoze_li0406@163.com (H.L.); myemail20231026@163.com (Y.W.); mabaolong88@163.com (B.M.); 2Hebei Province Key Laboratory of Intelligent Industrial Equipment Technology, Hebei Engineering University, Handan 056038, China; gaoruizhen@hebeu.edu.cn

**Keywords:** prestressed concrete cylinder pipe (PCCP), fiber-optic distributed acoustic sensor (DAS), YOLOv5, deep learning, wire-break monitoring

## Abstract

The failure accidents of prestressed concrete cylinder pipe (PCCP) seriously affect the economic feasibility of the construction site. The traditional method of needing to stop construction for pipe inspection is time-consuming and laborious. This paper studies the PCCP broken wire identification algorithm based on deep learning. A PCCP wire-breaking test platform was built; the Distributed Fiber Acoustic Sensing Monitoring System (DAS) monitors wire-breakage events in DN4000mm PCCPs buried underground. The collected broken wire signal creates a time-frequency spectrum diagram dataset of the simulated broken wire signal through continuous wavelet transform (CWT). Considering the location of equipment limitations, based on the YOLOv5 algorithm, a lightweight algorithm, YOLOv5-Break is proposed for broken wire monitoring. Firstly, MobileNetV3 is used to replace the YOLOv5 network backbone, and Dynamic Conv is used to replace Conv in C3 to reduce redundant computation and memory access; the coordinate attention mechanism is integrated into the C3 module to make the algorithm pay more attention to location information; at the same time, CIOU is replaced by Focal_EIoU to make the algorithm pay more attention to high-quality samples and balance the uneven problem of complex and easy examples. The YOLOv5-Break algorithm achieves a mAP of 97.72% on the self-built broken wire dataset, outperforming YOLOv8, YOLOv9, and YOLOv10. Notably, YOLOv5-Break reduces the model weight to 7.74 MB, 46.25% smaller than YOLOv5 and significantly lighter than YOLOv8s and YOLOv9s. With a computational cost of 8.3 GFLOPs, YOLOv5-Break is 71.0% and 78.5% more efficient than YOLOv8s and YOLOv9s. It can be seen that the lightweight algorithm YOLOv5-Break proposed in this article simplifies the algorithm without losing accuracy. Moreover, the lightweight algorithm does not require high hardware computing power and can be better arranged in the PCCP broken wire monitoring system.

## 1. Introduction

A prestressed concrete cylinder pipe (PCCP) is a composite piping material consisting of a concrete pipe core with a steel cylinder wound with a high-strength prestressed steel wire and a protective cement mortar layer [1]. As of 2024, 35,000 km of PCCP pipelines have been installed in North America [2]. They are widely used in long-distance, large-scale drainage projects because they can be prefabricated in factories, transported to the construction site for assembly quickly, and have the advantages of good sealing, pressure resistance, and low production cost. However, in the long-term working process especially, PCCP will inevitably be affected by the superimposed effects of the soil environment of the laying site, the hydrogen embrittlement of prestressed steel wire, external load, and other aspects, which will have different degrees of negative impact on the pipeline. The accumulation of negative impact will cause pipeline rupture, especially for the fracture of prestressed steel wire; with the increase in the number of broken wires, the bearing capacity of the section of PCCP will gradually decrease, and when the number or proportion of broken wires in a certain section of pipeline reaches a certain degree, there will be a risk of tube explosion, which may cause serious threats and major social hazards to the laying site and the surrounding environment [3,4]. PCCP compressive strength comes from the spiral winding in the prestressing steel wire on the steel cylinder concrete. Therefore, for the pipeline, prestressing steel wire damage or corrosion is the main factor of PCCP failure [5]. To deal with the safety risks caused by the erosion and damage of PCCP prestressed steel wires, it is necessary to effectively detect the potential safety hazards of the PCCP pipeline and evaluate and predict the degree of harm [6]. It has been shown that the energy released by the wire-break signal is transmitted through acoustic waves in water [7]. The central monitoring technologies include internal visual detection, echo detection, electromagnetic detection, acoustic detection, etc., but they are all limited by the problem of real-time monitoring efficiency. Due to the discontinuity of time and space in traditional hydraulic monitoring, it is difficult to find potential hidden dangers. It is necessary to continuously monitor for broken wire during the operation period to detect broken wire events, grasp the degree of the broken wire, and to provide an early warning and alarm to avoid PCCP tube explosion. Fiber-optic distributed acoustic sensors (DAS) are one of the most attractive and promising fiber-optic sensing technologies [8]. Therefore, DAS is widely used in long-term monitoring projects. Li et al. [9] showed that DAS can accurately locate sound sources and detect weak sound signals with high sensitivity, enabling the simultaneous monitoring of multiple parameters all while covering a large monitoring area and significantly reducing equipment usage. Therefore, DAS is widely used in long-term monitoring projects. Xu et al. [10] proposed a plan to monitor microcracks in cement mortar protective layers using Brillouin optical time domain analysis (BOTDA) strain sensors. Huang et al. [11] proposed a plan to monitor microcracks in cement mortar protective layers using Brillouin optical time domain analysis (BOTDA) strain sensors. Stajanca et al. [12] proposed a simple leakage detection method based on time-domain average DAS signal spectral integration, which detected gas pipeline leakage by winding sensing fiber directly on the pipeline. The results show that this method is feasible.

In recent years, scholars at home and abroad have also tried to apply depthology to analyze signals collected by distributed acoustic sensing systems. Combining DAS systems and an object detection algorithm offers multiple advantages. DAS systems continuously monitor acoustic signals along the entire length of the optical fiber, providing high-resolution and real-time information on environmental changes and covering long distances and vast areas. However, relying on DAS signal analysis alone may make it challenging to accomplish the task accurately. The target detector can deeply analyze the signal captured by DAS, identify the specific acoustic characteristics, and better complete the acoustic signal monitoring task.

CNN, a representative algorithm for deep learning, usually consists of an input layer, convolutional layer, pooling layer, fully connected layer, and output layer [13]. Different filters (i.e., convolutional kernels) extract feature information at different scales in the convolutional layer. Object detection methods based on CNN are classified into two categories based on detection speed: two- and one-stage object detection [14]. One-stage object detection algorithm performs object detection through a single forward pass, offering high speed and a simpler training process, making it suitable for real-time applications. However, it tends to have lower accuracy, especially when detecting small objects or in complex backgrounds. On the other hand, the two-stage object detection algorithm first generates candidate regions before performing classification and regression, achieving higher accuracy, particularly for complex scenes and small object detection. However, it involves higher computational cost, slower speed, and a more complex training process, making it more suitable for applications where high accuracy is required but real-time performance is less critical. The YOLO (You Only Look Once) algorithm series is among the best because of its excellent detection speed in one-stage object detection algorithms [15]. YOLOv5 is quickly gaining attention for its flexibility and good architecture. In the past two years, the YOLO series has been updated to YOLOV 10 [16,17,18,19].

Deep learning-based sound classification methods often convert audio files into image forms, such as Spectral images, and then use neural networks to process the images [20,21]. Zhang et al. [22] converted one-dimensional acoustic signals into time-frequency spectrograms and used convolutional neural networks to identify the image class attributes of the spectrograms to obtain a relatively ideal recognition accuracy, which was also the best accuracy reported at that time. In 2020, Peng et al. [23] proposed an integrated technical framework that combines distributed fiber sensing technology and CNN to identify external interference and pipeline defects with high accuracy. In the same year, Jalkampudi et al. [24] developed a CNN-based model for automatically detecting footstep signals in environmental seismic recordings in distributed acoustic sensing arrays in cities. Stork et al. [25] used the YOLO V3 model to process microseismic signals collected by distributed acoustic sensing systems with a precision that completely exceeds that of manual detection. Zhang et al. [26] proposed an improved YOLOv7 algorithm for the multi-event real-time detection of DAS systems, and the algorithm achieved a 99.7% recognition of six pipeline security event types. Ma et al. [27] proposed a PCCP broken wire automatic monitoring technology using a distributed optical fiber acoustic wave sensing monitoring system (DAS) and introduced the YOLOX algorithm to recognize broken wire signals, achieving excellent results on the self-built dataset. Han et al. [28] realized efficient intrusion detection and pulse event classification through telecommunication infrastructure, combined multi-task learning, neural network architecture, and edge computing, and significantly improved public safety and environmental monitoring capabilities. Li et al. [29] proposed the YOLO-TP network, specifically designed for tobacco beetle detection based on YOLOv8n. By optimizing the architecture and loss function, the model significantly reduces computational complexity and parameters while achieving a detection accuracy of 99.5%, providing an efficient solution for tobacco beetle detection. These studies show that deep learning technology based on convolutional neural networks can effectively build accurate models of broken wire and sound events. However, previous work has focused on pre-processing existing signals or training input models with perturbation signals, respectively, and the robustness of the models needs to be further verified.

Considering the long duration of wire-break monitoring events and the limited computing power of the monitoring point processor, this paper decides to adopt the YOLOv5 algorithm and combine the signals collected by the DAS system for verification. In this study, a broken wire monitoring test platform was built, a 1:1 PCCP prototype pipeline fracture monitoring test was designed, a distributed acoustic fiber system was deployed to monitor broken wire events, and an improved lightweight algorithm, YOLOv5-Break, was proposed for broken wire monitoring events. The broken-wire signal collected by the DAS system was combined with the recorded pipeline running noise to simulate the actual situation more accurately, and the three were used to generate a PCCP broken-wire spectrum dataset by CWT transformation and then used for YOLOv5-Break algorithm training to verify its feasibility.

## 2. Improved YOLOv5 Algorithm

### 2.1. YOLOv5

In this study, the improved algorithm selection is YOLOv5. The YOLOv5 network structure is divided into backbone, neck, and head. The input uses the Focus module with built-in Mosaic data enhancement and K-means algorithm to adaptively process the input image. The backbone part includes the Conv layer and four C3 modules to generate feature maps of different sizes. Finally, the Spatial Pyramid Pooling-Fast (SPPF) structure is used to fuse the receptive fields. The neck partial utilizes the Feature Pyramid Network and Path Aggregation Network, a combined path aggregation network architecture. The detection partially decodes different sizes of feature maps and uses the Non-Maximum Suppression (NMS) algorithm to obtain the optimal target prediction box and category position information. The YOLOv5 network structure is shown in Figure 1.

Considering the limitations of broken wire event monitoring, deploying a complex network algorithm with many parameters into the device will occupy a large amount of running space, and the detection speed will be slow. This makes it challenging to meet the real-time and rapid requirements of broken wire monitoring. Therefore, the YOLOv5 network structure needs to be improved. The improvement strategy focuses on lightweight design, and the following sections introduce algorithms for lightweight optimization.

### 2.2. MobileNetV3

MobileNetV3 [30], as a cutting-edge neural network architecture, has achieved significant success in computer vision. MobileNetV3 achieves excellent computing efficiency by adopting optimization strategies, such as depth-separable convolution, lightweight block design, and the Squeeze-and-Excitation (SE) attention mechanism. It is suitable for mobile devices, embedded systems, and resource-intensive systems and provides efficient performance in restricted environments. Figure 2 shows the MobileNetV3 structure diagram.

The YOLOv5 backbone uses traditional convolutions and C3 modules, focusing on balancing computational efficiency and high accuracy, making it suitable for environments with strong computational power. In contrast, MobileNetV3 employs HSwish activation and SE modules to reduce computational complexity and memory usage, making it suitable for resource-constrained devices. Therefore, in this study, MobileNetV3 is used to replace the YOLOv5 backbone to meet the requirements of portability, speed, and efficiency. While retaining the gradual downsampling and feature extraction process, the network depth is reduced, making the model simpler and more efficient.

### 2.3. Replacing Conv with Dynamic Convolution

Dynamic Conv [31] was proposed to solve the problem of lightweight convolutional neural networks with degraded performance and characterization ability. The dynamic convolution process is shown in Figure 3. The convolution kernel parameters of dynamic convolution can be dynamically adjusted during the training process to adapt to different types of input data, thereby improving the robustness of the algorithm. Therefore, dynamic convolution can better capture local features and global correlations and understand complex data structures.

Compared with traditional convolution, another significant advantage of Dynamic Conv is that the number of parameters is small. The parameters of Dynamic Conv are shareable, reducing the algorithm’s memory footprint and the risk of overfitting, thereby improving the algorithm’s generalization ability. The convolutional neural network formula is as follows:(1)$$y=g(W^{T}x+b)$$Dynamic convolutional neural networks can be defined as follows:(2)$$\begin{array}{l} y=g({\tilde{W}}^{T}(x)x+b(x))\\ \tilde{W}(x)=\sum_{k=1}^{K}\pi_{k}(x){\tilde{W}}_{k},\\ \tilde{b}(x)=\sum_{k=1}^{K}\pi_{k}(x){\tilde{b}}_{k}\\ 0\leq \pi_{k}(x)\leq 1,\sum_{k=1}^{K}\pi_{k}(x)=1 \end{array}$$

$\pi_{k}(x)$ represents the weight of the attention of the ${\tilde{W}}^{T}(x)x+b(x)$ linear function, which varies dynamically with the input *x*. Therefore, given a given input, dynamic convolution gives the best linear combination that fits that input.

In the improved model, all the convolution layers in the YOLOv5 neck section are replaced with dynamic convolutions to enhance the algorithm’s perception of the input data, allowing it to focus on important information and thereby improving the algorithm’s performance.

### 2.4. Adding Coordinate Attention

Coordinate attention (CA) [32] is an attention mechanism used to enhance the performance of deep neural networks. It aims to algorithmically weigh the correlation between different channels of feature maps in the input data. CA can automatically learn the importance of each channel and dynamically adjust the channel’s weight to better capture and utilize the correlation information between channels in the algorithm. CACA is shown in Figure 4.

The core idea of the CA is to use global pooling operations to concatenate the characteristics of each channel and generate a global channel weight. For the information of each channel of the input feature map, a Conv kernel of size sum is first used to perform a global average pooling operation on each channel’s X and Y directions; the features of the two spatial rules and the output direction-aware feature map are concatenated. The information feature map equation of C×H×1 in the X and Y is obtained as follows:(3)$$Z_{c}^{h}\left(h\right)=\frac{1}{W}\sum_{i=0}^{w}x_{c}\left(h,i\right),Z_{c}^{h}\in R^{C\times H\times 1}$$(4)$$Z_{c}^{w}\left(w\right)=\frac{1}{H}\sum_{i=0}^{h}x_{c}\left(j,w\right),Z_{c}^{w}\in R^{C\times 1\times w}$$

H represents the height component. *W* represents the width component. $x_{c}$ represents the channel of input *x*, and $x_{c}(h,i)$ represents the h component of each channel of input *x*. 

Afterward, the CA generation operation is performed. The generated feature maps $Z_{c}^{h}$ and $Z_{c}^{w}$ are cascaded along the spatial dimension, and the features in the X and Y directions are cascaded into global features. Then, a 1 × 1 Conv kernel is used to perform dimensionality reduction and $F^{1}$ activation operations to generate feature maps. Then, along the spatial dimension, the feature map is *Split* into $f^{h}$ and $f^{w}$. The $F^{1}$ activation operations are as follows:(5)$$f=\delta \left(F^{1}\left(\left[Z^{h},Z^{w}\right]\right)\right),f\in R^{C/r\times \left(H\times W\right)\times 1}$$

Finally, two 1 × 1 Conv, $F_{h}$ and $F_{w}$, are used to perform dimensionality enhancement operation and then combined with the Sigmoid activation function to obtain the final attention vectors $g^{h}$ and $g^{w}$:(6)$$g^{h}=\sigma \left(F_{h}\left(f^{h}\right)\right)$$(7)$$g^{w}=\sigma \left(F_{w}\left(f^{w}\right)\right)$$

Coordinate attention to the final expression is as follows:(8)$$y_{c}\left(i,j\right)=x_{c}(i,j)\times g_{c}^{h}\left(i\right)\times g_{c}^{w}\left(j\right)$$

By integrating coordinate attention into the C3 module, the number of network layers is not increased, and the required training parameters are also microscopic. In this study, the CBS module of the original YOLOv5 model was reconstructed. Based on the C3 module, the Bottleneck was replaced with CABottleneck to enhance the model’s functionality. CABottleneck incorporates attention mechanisms and reduces the depth of the network, enabling the C3_CA module to not only retain the main structure of C3 but also better focus on important channel information during feature processing, thereby improving performance. The results are shown in Table 1. The fusion process is illustrated in Figure 5.

### 2.5. IoU Loss Improvement

Intersection over Union (IoU) loss [33] is a standard indicator used to measure the performance of target detection tasks. IoU represents the intersection ratio concatenate and algorithm prediction areas. The IoU calculation equation is as follows:(9)$$IoU=\frac{Intersection}{Union},0<IoU<1$$

Intersection and Union represent the intersection and union of the above two areas. IoU loss is usually used for target detection performance evaluation, but it only reflects the degree of overlap of different prediction boxes. When two prediction boxes, A and B, do not intersect, IoU cannot represent the distance between them, and gradient backpropagation cannot be performed, making practical training impossible. In addition, when the IoU is the same, location information cannot be provided, leading to prediction boxes selection problems.

In the improved model, Efficient Intersection over Union (EIoU) [34] was introduced into the model. EIoU adds a penalty term of the detection scale based on IoU, separates the influencing factors of the aspect ratio of the predicted box and the ground truth box, and calculates the length and width of the expected box and the actual box. respectively. Through splitting, EIoU has faster convergence speed and better positioning results. The EIoU calculation equation is as follows:(10)$$\begin{array}{l} L_{EIoU}=L_{IoU}+L_{dis}+L_{asp}\\ =1-IoU+\frac{\rho^{2}\left(b,b^{gt}\right)}{c^{2}}+\frac{\rho^{2}\left(w,w^{gt}\right)}{C_{w}^{2}}+\frac{\rho^{2}\left(h,h^{gt}\right)}{C_{h}^{2}} \end{array}$$

$L_{IoU}$ represents IoU loss, $L_{dis}$ represents distance loss, and $L_{asp}$ represents height and width loss.

To pay better attention to higher-quality anchor boxes, Focal Loss is added to EIoU to make the loss function better pay attention to the imbalance of positive and negative samples. Focal Loss can not only suppress the impact of low-quality samples on the loss but also can, with the error rate at the time, adaptively adjust the gradient so that the slope is maintained within a stable range and does not fluctuate violently. The Focal_EIoU calculation equation is as follows:(11)$$L_{Focal\_EIoU}={IoU}^{\gamma }L_{EIoU}$$

### 2.6. YOLOv5 Improvement Results

The specific methods are as follows: (1) To make the network algorithm portable and able to detect quickly, the lightweight module MobileNetv3-small is used to reconstruct the backbone network of YOLOv5 and reduce the size and parameter amount of the algorithm. (2) In the neck part, dynamic convolution is introduced to replace the ordinary convolution in the C3 module of the network to reduce redundant calculations. (3) The CA module is added inside C3 to compensate for the accuracy loss caused by the lightweight algorithm. (4) The YOLOv5 algorithm loss function CIoU is replaced with Focal_EIoU to increase algorithm convergence speed and positioning accuracy. The improved network structure is shown in Figure 6.

## 3. Training and Data Processing

### 3.1. Wire-Break Monitoring Test

A time-domain Distributed Acoustic Sensing (DAS) system with a spatial resolution of 10 m was adopted in the wire-breaking test. The system model was BST-02D40, and the effective monitoring distance of the system was 40 km. The system architecture diagram is shown in Figure 7. The light source of the test system is a narrow linewidth low-frequency drift laser (Lazer). An acousto-optic modulator (AOM) modulates the continuous light waves from the laser to generate pulses. The modulated pulses are amplified by the first erbium-doped fiber amplifier (EDFA) and then transmitted to the sensing fiber, model GTA-4B1.3, via a halo manipulator (CIR). The Rayleigh scattered signal is amplified by a second EDFA for a better signal-to-noise ratio and injected into a port of the 3 × 3 coupler by a second circulator. Two Faraday rotating mirrors (FRMS) are connected to two ports on the other side of the coupler with an optical path difference of 5 m. The final interference information of the coupler output is collected by three photodetectors (PDS).

To simulate the conditions of PCCP pipelines during real-world operation and obtain wire-breakage signals close to reality, the test environment was constructed using four DN4000mm PCCPs with an internal diameter of 4 m and a total length of 20 m, which are buried underground. The schematic of the test setup is shown in Figure 8. The actual operating conditions of the pipeline were simulated by sealing the pipes and injecting water and pressurizing the pipelines with a pressure pump. Additionally, sensor arrangements were designed to be fixed on the inner wall of the pipeline and to float within the pipe.

Before the experiment, holes were drilled at both ends of the PCCP at the sealing plates, and a customized optical cable entry sealing device was installed at the inlet. The armored communication optical cable, which is pressure-resistant and waterproof, was laid from the ground control room, entering through the hole at the sealing plate and extending to the end of the pipeline before being looped back and exiting at the same location. The armored installation method for each experiment stage was consistent to ensure accurate signal detection. The gap between the optical cable and the inlet device was sealed with polyurethane. The optical cable section from the inlet to the end of the pipeline was fixed to the inner wall of the PCCP using water-resistant adhesive. In contrast, the returning optical cable was naturally placed in the water. After exiting the pipeline, the optical cable was adhered to the outer wall of the pipe using reserved adhesive.

The main optical cable and the acoustic optical fiber tail cables were fused, and after splicing the tail fibers in the equipment room, they were connected to the DAS instrument. The pipeline’s drain valve at the tail was closed, and the pipeline was filled with water. A pressure pump was used to pressurize the pipeline, maintaining a stable internal pressure. A pneumatic hammer and cutting machine were used to open a window in the outer layer of the pipe’s mortar, exposing the outer prestressed steel wires for the wire-break operation.

### 3.2. Broken Wire Data Processing

The data acquisition system in this experiment was a DAS-distributed optical fiber acoustic wave sensing monitoring system and a laptop computer. At the beginning of the test, the manual cutting operation was performed. In the cutting process, each steel wire under a window was uniformly cut at first but not directly cut, and then the incision was gradually expanded. When one or a few steel wires were broken, the cutting was stopped, and the steel wires with incisions around them were naturally broken due to increased stress. In order to ensure the safety of the test, the number of broken wires in each window was controlled at about 15. The test site is shown in Figure 9.

A distributed optical fiber acoustic wave sensor monitoring instrument was used to monitor the wire fracture eventa and save the field data, with an offline demodulation sampling interval of 0.05 ms, a collection time of 12 s, and the continuous recording of 240,000 data of each corresponding breaking point. This study uses Continue Wavelet Transform (CWT) [35] to process the collected broken wire signals to obtain images with transparent texture. The typical waveform of the broken wire signal is shown in Figure 10.

CWT excels in non-stationary signal processing and allows multi-scale analysis. Through wavelet basis functions at different scales, the details and overall characteristics of the signal can be observed simultaneously, and the signal’s transient, edge, and frequency changes can be captured. When analyzing signals, CWT can parallelize the time domain and frequency domain, allowing a more comprehensive understanding of the time-frequency characteristics of the signal. The CWT of signal *x* can be defined as follows:(12)$$CWT_{x}^{\psi }\left(\tau ,s\right)=\Psi_{x}^{\psi }\left(\tau ,s\right)=\frac{1}{\sqrt{s}}\int x\left(t\right)\psi \left(\frac{t-\tau }{s}\right)dt,s\neq 0$$

In the equation, x(t) represents the pure harmonic signal, ψ(t) defines the wavelet basis function. x(t) is defined as follows:(13)$$x(t)=A\cos\left(\omega t\right)$$

ψ(t) is a single wavelet. After translation or scaling transformation, a “wavelet” family can be obtained. This can be introduced into the following equation:(14)$$\psi_{s,\tau }(t)=\frac{1}{\sqrt{s}}\psi \left(\frac{t-\tau }{s}\right),s,\tau \in R,s\neq 0$$

In the equation, *s* is called the scale (stretching) parameter; *b* is called the displacement parameter. At this time, there is the following equation:(15)$${\left\|\psi_{s,\tau }(t)\right\|}^{2}=\frac{1}{s}\int_{-\infty }^{+\infty }\left|\psi {\left(\frac{t-\tau }{s}\right)}^{2}\right|dt=\int_{-\infty }^{+\infty }\left|\psi {\left(t\right)}^{2}\right|dt={\left\|\psi (t)\right\|}^{2}$$ψ(t) is translated or stretched to obtain $\psi_{s,\tau }(t)$, so it is also called the mother wavelet.

The CWT can also be rewritten as a convolution expression:(16)$$\Psi_{x}^{\psi }\left(\tau ,s\right)=\frac{1}{\sqrt{s}}\int x\left(t\right)\psi \left(\frac{t-\tau }{s}\right)dt=\frac{1}{\sqrt{s}}x(t)*{\bar{\psi }}_{s}(t),s>0$$(17)$${\bar{\psi }}_{s}(t)=\frac{1}{s}\bar{\psi }\left(-\frac{t}{s}\right)$$
∗ represents convolution operations.

### 3.3. Broken Wire Dataset

The DAS system collected 60 1:1 prototype wire-break test pure wire-break signals. Firstly, the 60 pure sound signals of broken wire were coded as 1~60, and the synthetic signals were combined with the background noise of the pipeline; 300 wire-breaking signals simulating the actual working conditions of the pipeline were obtained. The typical combination of broken wire signal and pipeline operation noise is shown in Figure 11. The pipeline background noise was through hydrophones recorded under the operational conditions of a PCCP. This approach ensures that the signals are more representative of real-world conditions, thereby imposing higher demands on the algorithm’s noise interference resistance.

Then, continuous wavelet transform (CWT) is used to obtain the time-frequency graphs of 300 signals, as shown in Figure 12. The characteristic value of the broken wire signal is mainly concentrated below 3.5 kHz, and the duration of the broken wire energy release is about 0.05 s. Therefore, this region has also become the main area identified in subsequent studies.

To enrich the dataset, we used a Geode 24-channel seismometer and land-towed geophone system to simulate the pipeline vibration pulse signal caused by broken wire using the pulse vibration signal generated by the hammer to the ground. The signal was collected through the hydrophone to obtain 150 vibration signals, as shown in Figure 13.

Then, the mixed broken signal is used as region A, and the pure noise is used as region B to input the GAN network. The false broken wire synthesis spectrum is obtained: The data of 300 simulated broken wires were expanded to 1600, while 1750 broken wire datasets were obtained by mixing with vibration signals. According to the ratio of 8:2, the training set and verification set are obtained and used for model training, and the broken wire feature locations were manually marked, as shown in Figure 14.

### 3.4. Network Training

#### 3.4.1. Training Parameters and Evaluation Indicators

After cropping the images of the input algorithm to a size of 640 × 480, two classes were set. The batch size was set to 16, the optimizer selected the stochastic gradient descent (SGD) to update network parameters, and the learning rate was set to 0.01; the network parameters were dynamically adjusting using a cosine learning rate reduction strategy. The IoU threshold was set to 0.45, with the training epochs set to 150. To measure the real-time performance and portability of the algorithm, the algorithm detection performance evaluation indicators used in this study are mAP50, mAP50:95, weight size, F1 score, and giga floating point calculation amount (FLOPs/G).

#### 3.4.2. Ablation Study

By using the self-built dataset for the wire-break monitoring test, we verified whether each module proposed in this article is effective and compared the wire-break monitoring experiments of each improved network. The training environment uses the Windows 11 operating system. The hardware configuration is as follows: CPU: Intel(R) Core(TM) i7-8750H CPU @ 2.20 GHz; GPU: NVIDIA GeForce GTX 1050, 4096MiB. The programming environment is PyCharm 2023, and the deep learning framework is PyTorch.

Table 1 shows the results. The MobileNetV3 module was used to replace the YOLOv5 algorithm backbone. The results of test 1 show that P and mAP increased by 2.23% and 0.15%, respectively, indicating that MobileNetV3 turned the network structure lightweight without causing a loss of detection accuracy. The test 2 experiment further integrated the C3_Dynamic-Conv module. P, R, and mAP did not change significantly, but FLOPs/G dropped by 3.3, indicating that the C3_Dynamic-Conv module successfully replaced complex convolutions with dynamic convolutions with lower computational costs, dramatically reducing training costs while maintaining high performance. Test 3 replaces the C3_Dynamic-Conv module on the basis of YOLOv5 and adds the CA mechanism. It can be seen that P and R increase by 1.58% and 1.27%, respectively, and FLOPs/G also decrease, indicating that the C3_Dynamic-Conv module reduces computing costs and CA. This addition can effectively enhance the algorithm’s anti-interference ability. Test 4 is the final improved algorithm YOLOv5-Break. Compared with the experimental 2 FLOPs/G, there is no significant change, but P and R increase by 0.15% and 0.16, respectively, indicating that the improved algorithm is lightweight and has strong anti-interference ability. Compared with the original YOLOv5 network, YOLOv5-Break’s P, R, and mAP increased by 2.34%, 0.70%, and 0.42%, respectively, and FLOPs/G decreased by 7.7. It reduced the complexity of the algorithm and improved the detection accuracy, which satisfies the need for disconnections and real-time and rapid monitoring. The YOLOv5-Break algorithm has a lower number of parameters, which is more conducive to deployment on edge devices, reduces hardware costs, and is more relevant to the actual needs for broken wire monitoring.

## 4. Results and Analysis

To further verify the rationality and effectiveness of the algorithm improvement in this article, with all parameter indicators unchanged, the training improvement algorithm is compared with the current mainstream target detection algorithms YOLOv3-tiny, YOLOv5, YOLOv7-tiny, YOLOv8s, YOLOv9s, and YOLOv10s algorithms. The experimental results of the experiment are shown in Table 2.

It can be seen from the data in Table 2 that the mAP of YOLOv5-Break has not changed much compared to the YOLOv3-tiny, YOLOv5, YOLOv7-tiny, YOLOv8s, YOLOv9s, and YOLOv10s algorithms, reaching 97.72%. In addition, with an F1 score of 0.9651, its weight size is only 7.74 MB, and its computational complexity is 8.3 GFLOPs. Compared to YOLOv8s and YOLOv3-tiny, the weights are reduced by 65.6% and 53.37%, respectively, and the computational complexity is lowered by 71.0% and 36.15%. Compared to YOLOv7-tiny and YOLOv10s, YOLOv5-Break demonstrates absolute superiority in accuracy and efficiency. It also outperforms YOLOv9s with a 0.88% increase in mAP, a 61.9% reduction in weights, and a 78.5% decrease in computational complexity. Overall, YOLOv5-Break achieves an optimal balance between accuracy, model size, and computational complexity, making it an efficient solution for resource-constrained and real-time applications. To sum up, it can be seen that the YOLOv5-Break algorithm is more beneficial to the actual needs of broken wire monitoring than the current mainstream target detection algorithms.

The P and R curve results of the network before and after optimization are shown in Figure 15a,b, where (a) is the comparison of P curve results between the YOLOv5-Break algorithm and the YOLOv5 algorithm, and (b) is the comparison of R curve results. It can be seen that the overall detection accuracy of the optimized network is increased by 0.2%. The number of parameters of YOLOv5-Break algorithm is small, the P and R curves are not much different from the original algorithm, and the recognition of vibration signals is better than the original algorithm. Figure 16 shows the loss curve during the training process of the YOLOv5-Break algorithm. The algorithm converges after 150 rounds of iterations. Figure 17 shows the confusion matrix; this model can distinguish well between broken wires and vibration signals.

Figure 18 is a comparison of the detection results of YOLOv5 and YOLOv5-Break. It can be seen that the detection accuracy of YOLOv5-Break is preserved when the number of parameters is reduced by 50%, proving that its performance in PCCP broken wire monitoring events is better than YOLOv5.

Figure 19 shows the sample of detection errors. From positions 1 and 2 in the figure, it can be seen that when the color of the energy generated by the superposition of the vibration signal and noise is closer to the sample of the broken wire signal, detection errors are made by the model, which are marked as broken wire signals while also being marked as vibrations. In fact, it is the only superposition of vibration signal and noise. Position 3 and position 4 should be lighter in color and close to the noise spectrum, resulting in a low detection rate and missing detection. It can be observed that the model exhibits a high probability of false positives in the samples. The main focus of our future work will be to increase the classification threshold in order to reduce false positives. Additionally, optimizing evaluation metrics such as precision and F1 score will help to assess the impact of false positives. Finally, through cross-validation and repeated training and validation, the model will be continuously optimized to find the best balance between reducing false positives and improving overall performance.

## 5. Conclusions

In this study, we conducted a 1:1 prototype wire-breakage monitoring experiment. Simulated wire-breakage signals collected using the DAS system were transformed into time-frequency spectrograms via continuous wavelet transform, and a lightweight improved algorithm, YOLOv5-Break, was proposed to identify wire-breakage events.

The YOLOv5-Break algorithm achieved a mAP of 97.72% on the self-built wire-breakage dataset, outperforming YOLOv8, YOLOv9, and YOLOv10. Notably, YOLOv5-Break reduced the model weight to 7.74 MB, 46.25% smaller than YOLOv5. The computational cost of YOLOv5-Break is 8.3 GFLOPs, with efficiency improvements of 71.0% and 78.5% compared to YOLOv8s and YOLOv9s, respectively. This demonstrates that the proposed lightweight algorithm YOLOv5-Break simplifies the model without compromising accuracy and requires minimal hardware computing power. The recognition accuracy of the broken wire event reached 97.2%. Meanwhile, the shortcomings of the model in recognizing challenging samples have been noted, and this will be the primary focus of our future work. YOLOv5-Break demonstrates strong robustness in the broken wire detection task, with 97.72% mAP, 59.14% mAP_0.5:0.95%, and 8.3 GFLOPS of high computational efficiency. Although there is room for improvement in handling complex backgrounds and overlapping targets, its overall robustness is strong.

In future studies, we will conduct additional wire-breakage experiments, including samples with varying positions and numbers of broken wires. Firstly, more experiments will be carried out on samples with overlapping energy regions. Secondly, further algorithmic improvements will be made to reduce the false positive and false negative rates of hard-to-identify samples while maintaining the current performance. Finally, we plan to deploy the DAS equipment on operational PCCP pipelines, conduct real-world monitoring trials, and strive to collect actual wire-break data.

## Figures and Tables

**Figure 1 sensors-25-00977-f001:**
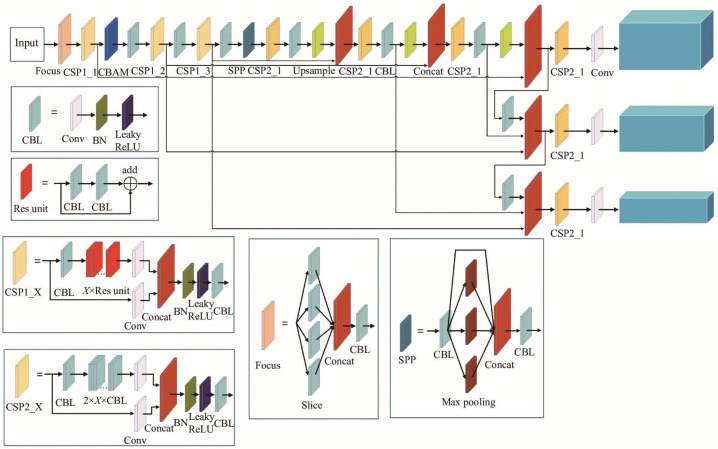
YOLOv5 network structure.

**Figure 2 sensors-25-00977-f002:**
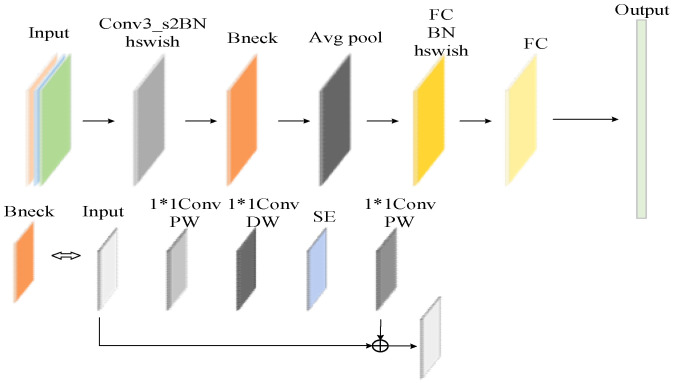
MobileNetV3 network structure.

**Figure 3 sensors-25-00977-f003:**
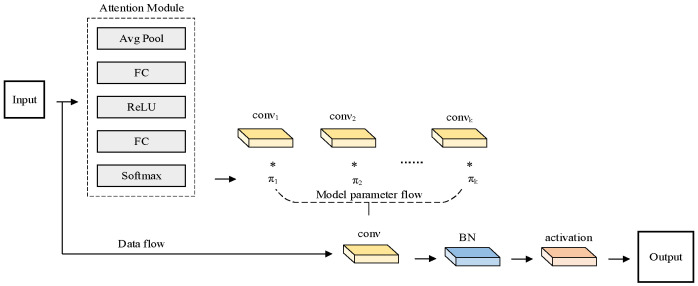
Dynamic Conv.

**Figure 4 sensors-25-00977-f004:**
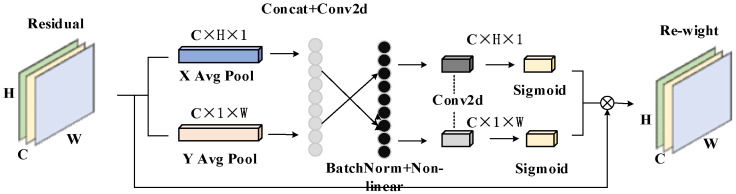
Coordinate attention [32].

**Figure 5 sensors-25-00977-f005:**
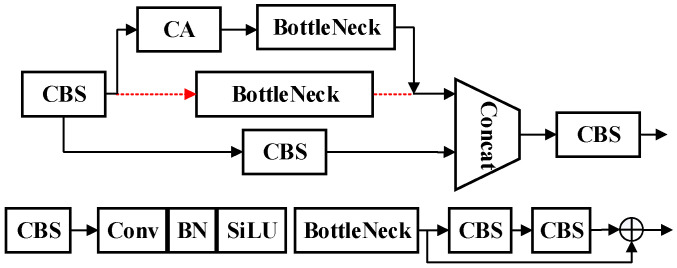
Fusion process.

**Figure 6 sensors-25-00977-f006:**
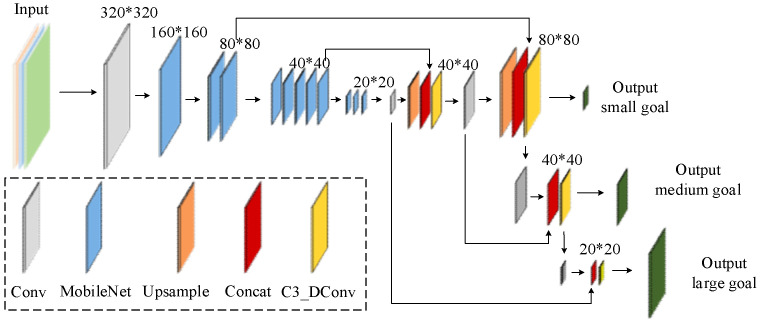
Improved network.

**Figure 7 sensors-25-00977-f007:**
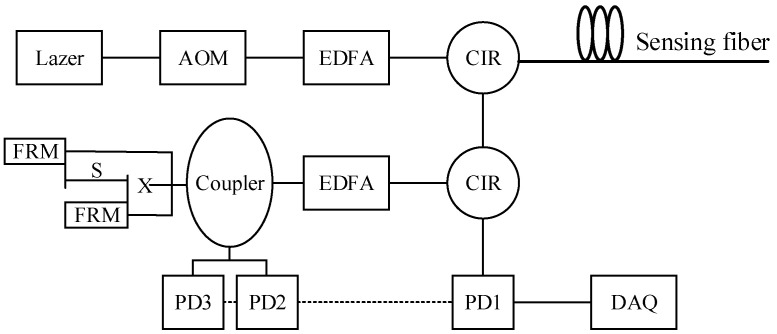
DAS experimental system settings.

**Figure 8 sensors-25-00977-f008:**
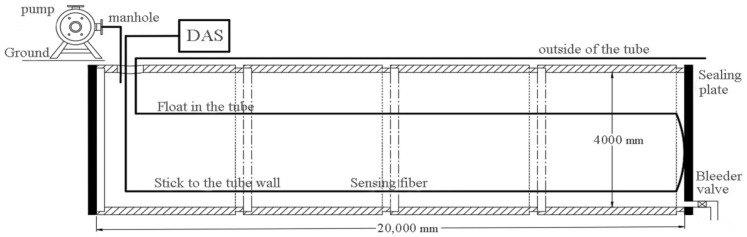
PCCP experimental device diagram.

**Figure 9 sensors-25-00977-f009:**
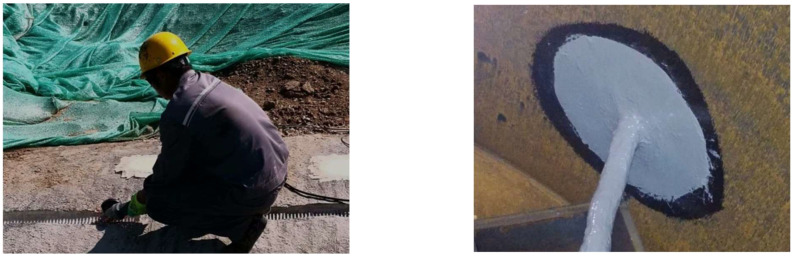
Test-site wire cutting.

**Figure 10 sensors-25-00977-f010:**
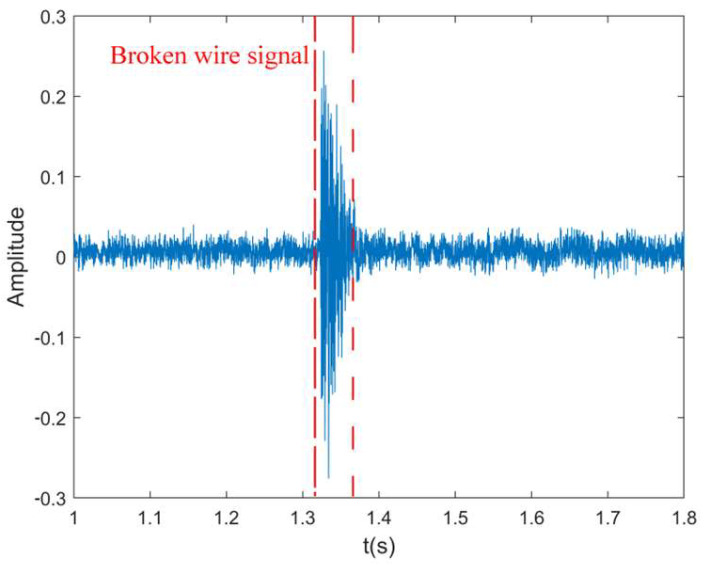
Typical broken wire waveform.

**Figure 11 sensors-25-00977-f011:**
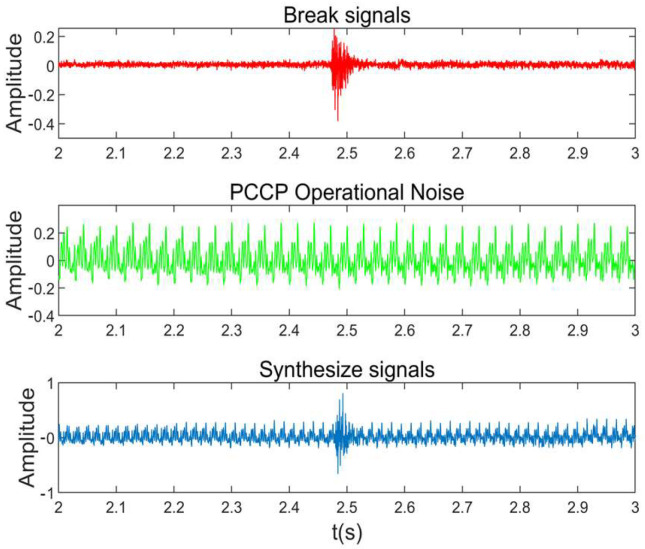
Typical broken wire fusion noise signal.

**Figure 12 sensors-25-00977-f012:**
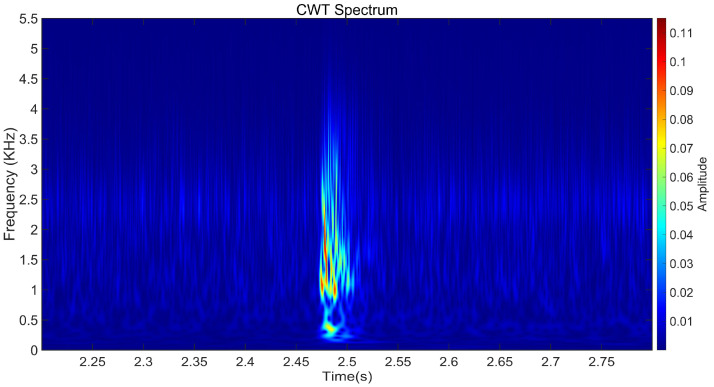
Synthesize broken wire signal spectrum transformed by CWT.

**Figure 13 sensors-25-00977-f013:**
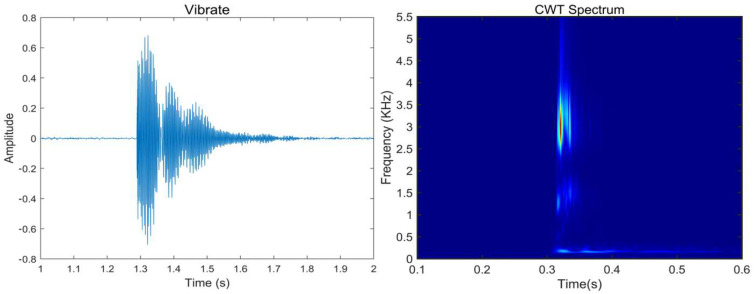
Vibration signals: time domain and frequency domain.

**Figure 14 sensors-25-00977-f014:**
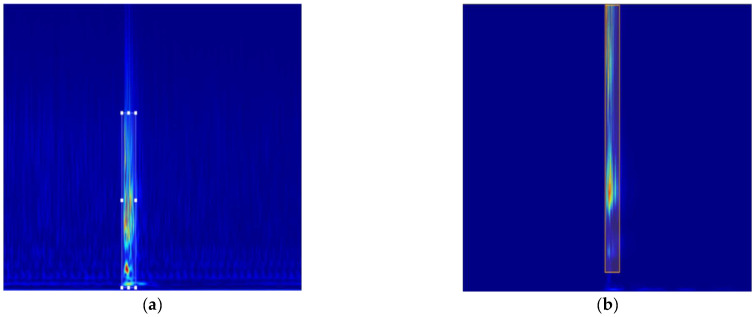
Wire-break spectrogram dataset. (**a**) Wire break signal; (**b**) vibration signal.

**Figure 15 sensors-25-00977-f015:**
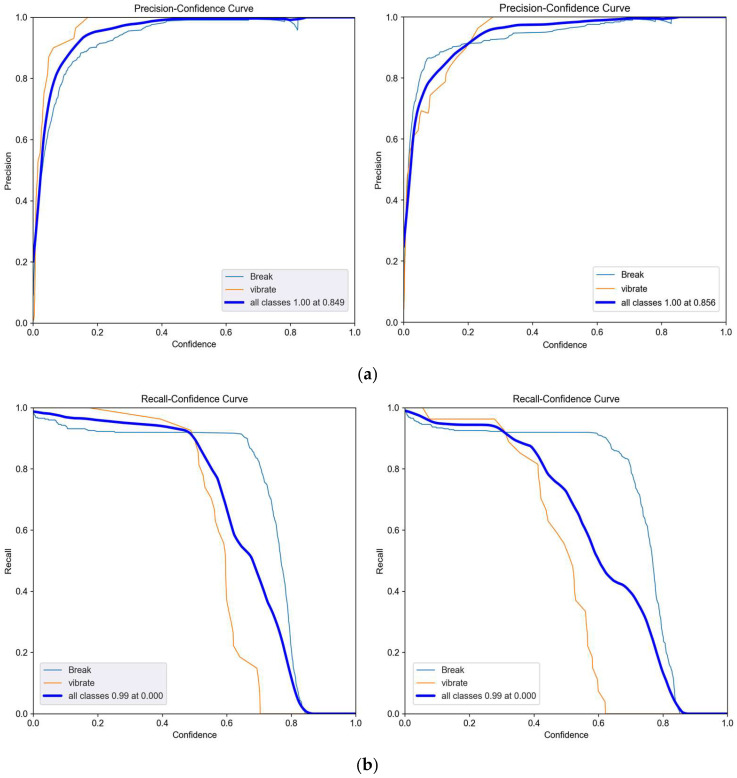
P and R Curve. (**a**) P-Curve contrast; (**b**) R-Curve contrast.

**Figure 16 sensors-25-00977-f016:**
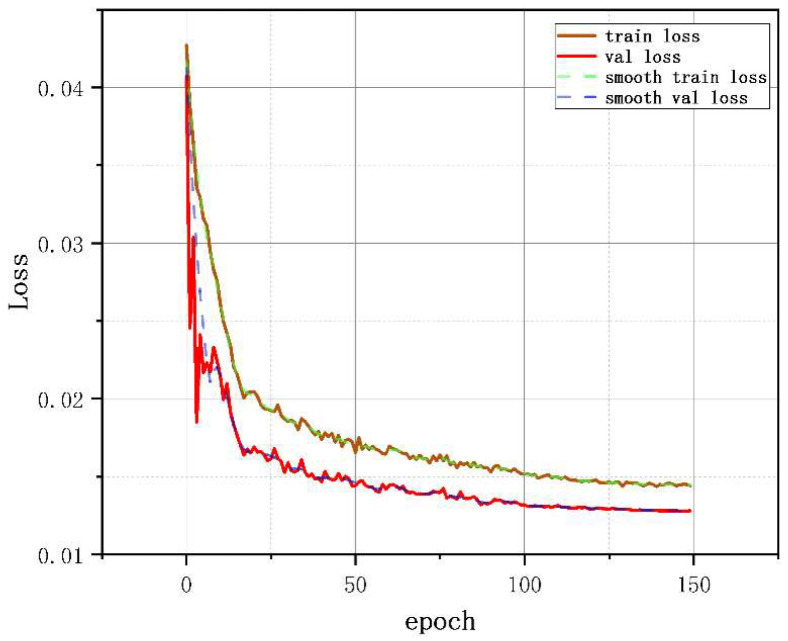
Loss curve.

**Figure 17 sensors-25-00977-f017:**
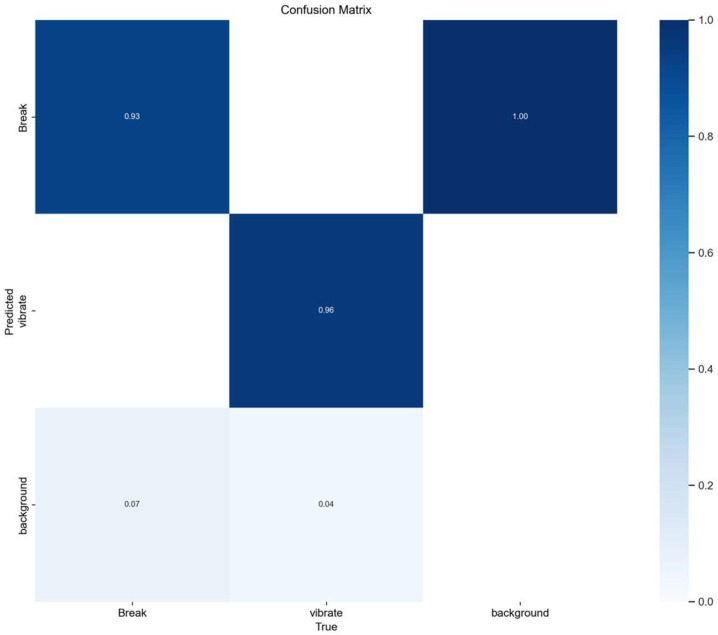
Confusion matrix.

**Figure 18 sensors-25-00977-f018:**
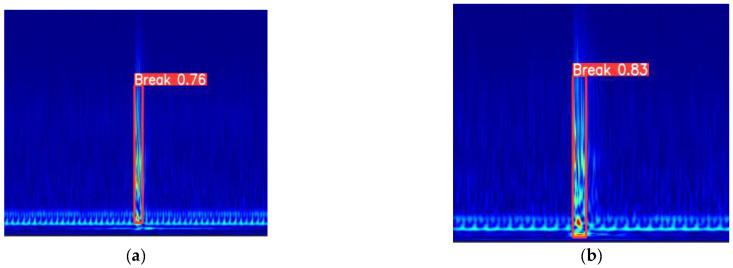
Test result example: (**a**) YOLOv5 detect results; (**b**) YOLOv5-Break detect results.

**Figure 19 sensors-25-00977-f019:**
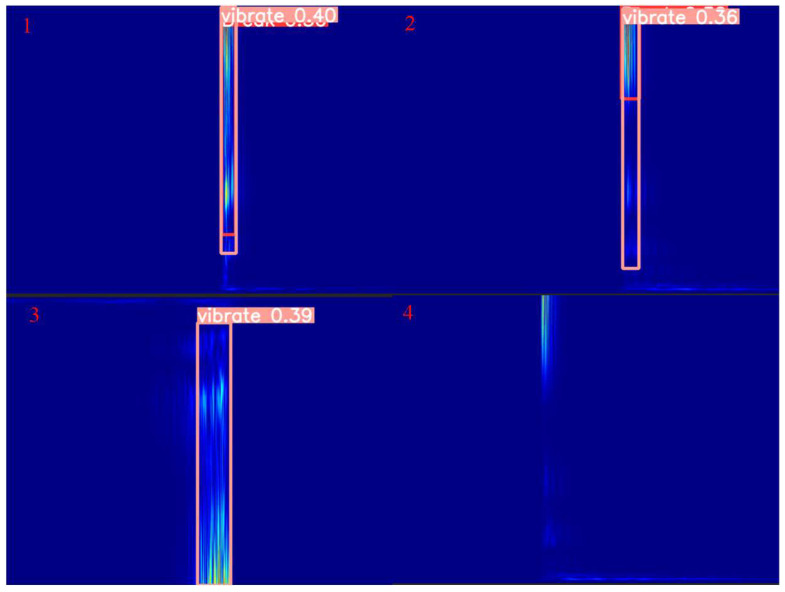
False drop pattern.

**Table 1 sensors-25-00977-t001:** Ablation study.

	MobileNetV3	C3_DConv	CA	P/%	R/%	FLOPs/G	mAP/%	mAP_0.5:0.95%
YOLOv5				96.61	94.56	16.0	97.30	0.55104
1	√			98.84	95.51	11.6	97.45	0.62906
2	√	√		98.80	94.65	8.3	97.38	0.56063
3		√	√	98.19	95.83	13.8	95.89	0.55624
4	√	√	√	98.95	95.26	8.3	97.72	0.59138

**Table 2 sensors-25-00977-t002:** Comparison of performance of different algorithms.

Algorithms	mAP/%	mAP_0.5:0.95%	Weight Size(MB)	F1 Score	FLOPS/G
YOLOv3-tiny	97.87	59.23	16.6	0.9655	13.0
YOLOv5	97.20	55.10	14.4	0.9508	16.0
YOLOv7-tiny	50.88	25.68	11.6	0.6225	13.2
YOLOv8s	97.62	64.13	22.5	0.9668	28.6
YOLOv9s	96.84	66.25	20.3	0.9394	38.7
YOLOv10s	54.37	23.13	16.5	0.6122	24.4
YOLOv5-Break	97.72	59.14	7.74	0.9651	8.3

## Data Availability

Data are contained within the article.

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
