# Peer review of "A Prestressed Concrete Cylinder Pipe Broken Wire Detection Algorithm Based on Improved YOLOv5"

_sensors, 2025, doi:10.3390/s25030977_

Round 1

Reviewer 1 Report

Comments and Suggestions for Authors

The paper proposes an algorithm called YOLOv5-Break, which is a modification of YOLOv5, to monitor and detect wire breakage events in prestressed concrete pipes (PCCP). This combines fiber optic distributed acoustic monitoring system (DAS) with deep learning techniques to accurately identify wire breakage events. After a detailed analysis of it, it needs improvements, which I present below, as well as questions that the authors must answer:

1 - The introduction or addition of a new chapter should detail similar studies in recent literature, especially in the context of modifications to YOLO algorithms and similar works. Without this information, it is not possible to justify the choices and contributions of this work.

2 - What are the limitations of this work? If so which ones?

3 - The description of the methods has gaps that must be improved.

4 - Authors must provide more details about the division of the dataset (training and validation) and the balancing of classes.

5 - Why don't the authors use other relevant metrics, such as mAP@0.5:0.95, Recall and Precision? They should include this in the article.

6 - Has the model been validated for scenarios with more severe noise or variable conditions? If not, why?

7 - The results presented are limited to some metrics such as mAP@0.5 and FLOPs. I recommend adding confusion matrix to highlight false positives and false negatives.

8 - Authors must include in the article a discussion of error rates (false positives and negatives), analyzing specific scenarios in which the model failed.

9 - They should include comparative graphs, such as Precision-Recall and ROC curves, to help evaluate overall performance.

10 - In my opinion, the conclusions should be more aligned with the results presented. They claim robustness and efficiency without providing complete evidence to support these claims.

11 - Authors should include more specific future improvements, such as, for example, evaluation in large-scale scenarios and real conditions and tests on edge devices with more restricted hardware.

12 - The references are coherent, but should be complemented with more recent studies, especially on advances in YOLO algorithms and their applications in structural monitoring scenarios.

Comments on the Quality of English Language

The English could be improved to more clearly express the research.

Author Response

Comments 1:  The introduction or addition of a new chapter should detail similar studies in recent literature, especially in the context of modifications to YOLO algorithms and similar works. Without this information, it is not possible to justify the choices and contributions of this work.

Response 1: Thank you for pointing this out. We agree with this comment. Therefore, we have added introductions to similar articles and improved our research.[ In the article, we reorganized the writing of the introduction, and added the recent literature in lines 103-131, which is of great significance to our work.]

Comments 2: What are the limitations of this work? If so which ones?

Response 2: We agree to your proposal. We have, accordingly, revised the conclusions to emphasize this point. [In the article, we have added the description of the defects of the paper to the conclusion. We have not described the defects in this aspect before, thank you for pointing out. The specific changes are in lines 494-501 of the paper.]

Comments 3: The description of the methods has gaps that must be improved.

Response 3: Thank you for pointing this out. We have, accordingly, revised the description of the methods. [In this paper, we have improved the description of the method, especially the experimental process and the division of data sets. Thank you very much for your points. The specific changes are in lines 145-512 of the paper.]

Comments 4: Authors must provide more details about the division of the dataset (training and validation) and the balancing of classes.

Response 4: We agree to your proposal. We have, accordingly, revised the division of the dataset (training and validation) and the balancing of classes. [Thank you for pointing out that in the article, we updated the details of data set production to facilitate readers' understanding, with specific improvements in lines 359-393.]

Comments 5: Why don't the authors use other relevant metrics, such as mAP@0.5:0.95, Recall and Precision? They should include this in the article.

Response 5: Thank you for pointing this out. We have, accordingly, revised the data analysis of the trials. [Thank you very much for your correction. We have added the index mAP@0.5:0.95 into the result comparison table and made specific improvements in table 1 and table 2.(411-412,457-458)]

Comments 6: Has the model been validated for scenarios with more severe noise or variable conditions? If not, why?

Response 6: The problem you pointed out helped us a lot. We should verify the performance of the model under more severe noise or variable conditions, but due to laboratory conditions and the end of the project, we could not proceed to the next step. In addition, the probability of the actual PCCP pipeline failure event is low, and we can not monitor it in a short time. Thank you very much for pointing out our shortcomings.

Comments 7: The results presented are limited to some metrics such as mAP@0.5 and FLOPs. I recommend adding confusion matrix to highlight false positives and false negatives.

Response 7: The problem you pointed out helped us a lot. In this paper, we add confusion matrix to further demonstrate the performance of our algorithm [Thank you very much for your correction. The detailed changes are on pages 472-473]

Comments 8: Authors must include in the article a discussion of error rates (false positives and negatives), analyzing specific scenarios in which the model failed.

Response 8: We agree to your proposal. Therefore, we have increased the display and analysis of false samples [In the article, we have added the analysis and display of false detection samples. Thank you very much for pointing out our problems, which is very helpful to us. The specific changes are in lines 482-491]

Comments 9: They should include comparative graphs, such as Precision-Recall and ROC curves, to help evaluate overall performance.

Response 9: Your comments are very helpful to us. Thank you very much. We have to make changes in the paper. [The specific improvement in the article is in lines 459-470]

Comments 10:  In my opinion, the conclusions should be more aligned with the results presented. They claim robustness and efficiency without providing complete evidence to support these claims.

Response 10: Thank you for pointing out our problem, which is very helpful to our research. We will further standardize the language to achieve rigorous science, and we have deleted this paragraph.

Comments 11: Authors should include more specific future improvements, such as, for example, evaluation in large-scale scenarios and real conditions and tests on edge devices with more restricted hardware.

Response 11:  Thank you very much for your valuable suggestions to us. Your suggestions are of great significance to our follow-up research. According to your suggestions, we have explained the future research direction at the end of the paper. [Thank you very much for your correction. The detailed changes are in lines 492-512]

Comments 12: The references are coherent, but should be complemented with more recent studies, especially on advances in YOLO algorithms and their applications in structural monitoring scenarios.

Response 12: Your suggestion is of great help to us. We have supplemented the latest research with your suggestions. [Thank you very much for your correction. The detailed changes are in lines 103-131]

Reviewer 2 Report

Comments and Suggestions for Authors

This paper is too simple. The innovation is insufficient, and the contribution is not outstanding. It is not suitable for publication. Papers can be improved in the following aspects:

1. The work as the presented content is too small, more data should be used for analysis and validation.

2. The beneficial effects described in the paper due to changes in artificial neural network architecture have not been proven.

Comments on the Quality of English Language

The English of this paper needs moderate revision.

Author Response

Comments 1: The work as the presented content is too small, more data should be used for analysis and validation.

Response 1:

Dear Reviewer,

Thank you for reviewing our paper and providing valuable feedback. We have made revisions based on your suggestions.

We have improved the structure of the paper according to your advice. The sections have been reorganized to ensure stronger logical flow and clearer structure. The related changes are marked in the submitted manuscript. Additionally, we have comprehensively supplemented the paper, particularly in the areas of data processing, dataset division, and performance analysis and validation, to better reflect the contributions of our work.

Thank you again for your help and guidance on our work.

Reviewer 3 Report

Comments and Suggestions for Authors

Interesting work; but you need to work on your English and Structure.

Additionally, I would suggest you to compare against algorithms which have been used for the same tasks.

Comments on the Quality of English Language

-        The work is good, but you need to work on the structure of the paper.

-        You also need to improve the written English;

-        Please do not use first person when writing a journal paper. Use passive instead.

Author Response

Comments 1: Interesting work; but you need to work on your English and Structure.

Additionally, I would suggest you to compare against algorithms which have been used for the same tasks.

Response 1: Very pleased to write your review, we have changed it item by item. We have taken your suggestions seriously and have made all improvements to prestressed concrete cylinder pipe.

Abstract: We have rewritten the summary and added the relevant content in accordance with your suggestions.

Introduction:

Line35: We have corrected it, thank you for pointing it out

Line41-43: Thank you very much for your suggestion, we have revised the introduction.

Line69: Thank you very much for your suggestion, we have made a new description of the study of Ma et al.

Line 83-85: Thanks for your suggestion, we have added a new description of their advantages and disadvantages. (The specific changes are in lines 96-104)

Line 87-88: Thanks for your suggestion, our research is more based on the application of computing power limited devices, so we have not made a comparison of the latest YOLO series. Thank you very much for your suggestion, which is the motivation for our follow-up study.

Line 90: We are very sorry to cause you trouble because of our careless language. We have changed it to "wire broken monitoring".

Line 89-92 :Thank you for pointing out that the core factor for us to choose yolov5 is its fast speed and relative stability.

Line 92:Thank you for your suggestion. It is the broken wire signal collected by DAS system.

Line93 :Thank you for your suggestion, it is the target detection algorithm, sorry for causing you confusion because of our language expression. Your suggestion will be adopted after consideration. Your suggestion will be of great help to us.

Improved YOLOv5s Algorithm

Thank you very much for your suggestions and corrections to our research, we have adopted all your suggestions and made changes. Thank you for your valuable help and feedback on our work. Your review has been of great significance to us, highlighting areas where we can improve. We have addressed all of your suggestions and made appropriate adjustments to the structure of the paper. Your recommendations have guided our improvements. We have made substantial revisions to the entire manuscript, correcting all the details you pointed out, and these changes are reflected in the paper. We have made every effort to improve the manuscript, with some changes marked in red; however, these modifications do not affect the content or structure of the paper. We sincerely appreciate your dedicated work and hope that the revisions will be well-received. Once again, thank you for your insightful comments and suggestions.

Training and Data Processing

Thank you for your comments, we have made changes. You highlighted structural issues in our paper, and we have made revisions based on your suggestions. We have separated the methodology and experiments. In Chapter 3, Section 3.1 describes the experimental process, Section 3.2 covers data collection and processing, Section 3.3 focuses on dataset preparation, and Section 3.4 details the network training. The revised structure is reflected in the updated manuscript, with all changes marked in red. Additionally, we have addressed the issues you pointed out in the figures, including fixing the text within the images. Thank you for your valuable feedback, which has greatly improved the quality of our paper.

Results and Analysis:

Thank you for reviewing our manuscript and providing valuable feedback. Your suggestions helped us identify shortcomings in our work and provided us with important directions for improvement. We have carefully addressed all the issues you raised and made the corresponding revisions.

We have adjusted the structure of the paper based on your suggestions, incorporating some sections into the "Results and Analysis" chapter to ensure clearer logic and a more reasonable structure. Following your advice, we have conducted a more in-depth discussion in the "Results and Analysis" section, particularly focusing on the significance of the experimental results and their implications for practical applications.

Figures 13 and 14 have been redrawn to improve clarity, and the text within them has been optimized for better presentation. We apologize for the confusion caused by our terminology. YOLOv5s-v7.0 is equivalent to YOLOv5, and we have updated this throughout the paper. Additionally, we replaced "P-R curve" with "Precision-Recall curve" to provide a clearer comparison of results.

In line with your suggestions, we also adjusted the order of discussions on training and testing, ensuring that training discussions now precede testing.

Once again, we sincerely thank you for your assistance and guidance. Your detailed review has significantly enhanced the quality of our paper. We greatly appreciate your time and effort, and we hope that this round of revisions meets your approval!

Conclusion:

Thank you for your help and guidance on our work. We have revised the conclusion section and hope it meets your approval.

4. Response to Comments on the Quality of English Language

Point 1:
         The work is good, but you need to work on the structure of the paper.

 You also need to improve the written English;

Please do not use first person when writing a journal paper. Use passive instead.

Response 1:

Dear Reviewer,

Thank you for your thorough review and valuable feedback on our paper. We have addressed each of your suggestions and made the necessary revisions.

In response to your comments, we have improved the structure of the paper. We have reorganized the sections to ensure stronger logical flow and clearer structure. The relevant changes have been highlighted in the submitted manuscript. Additionally, we have thoroughly revised the language throughout the paper, particularly in terms of expression and grammar, to ensure the English is more fluent and formal. We have made corrections to several language issues to enhance the overall readability.

As per your suggestion, we have changed all first-person expressions to passive voice, in line with the academic writing conventions. These revisions have been fully incorporated into the new version. Thank you again for your assistance and constructive feedback.

Reviewer 4 Report

Comments and Suggestions for Authors

1.The paper does not elaborate deeply enough on the seriousness of the broken wire problem of PCCP (prestressed concrete cylinder bobbin) and the limitations of existing monitoring technology, and fails to fully demonstrate the urgency and importance of the research. It is recommended to add a detailed background on the problem of PCCP wire breakage, including its impact on public safety, economic losses and other aspects, as well as the limitations of current monitoring techniques, so as to enhance the practical significance of the study.

2.Some charts in the paper have some problems such as unclear marks and irregular typesetting, which affect the information transmission effect of the charts. It is suggested to strengthen the standardization and professional requirements for chart production, and ensure that the chart is clearly marked and neatly formatted, so that readers can accurately understand the information of the chart.

3.When introducing the algorithm improvement, the paper fails to elaborate the specific methods, key technologies and principles of the improvement, which makes it difficult for readers to understand and evaluate the effect of the improvement. It is suggested to add a detailed description and explanation of the algorithm improvement method, including the improved ideas, methods, principles, etc., so that readers can deeply understand and evaluate the effect of the improvement.

4. PCCP broken wire detection needs to draw on the latest research trends in machine learning, multitasking and multimodal machine learning algorithms. Some machine learning algorithms based on multitasking and multimodality have good reference value for this task, such as YOLO-TP: A lightweight model for individual counting of Lasioderma serricorne. . Multi-task learning for hand heat trace time estimation and identity recognition, Deep soft threshold feature separation network for infrared handprint identity recognition and time estimation. . Meanwhile, the author should more effectively describe the practical application scope and significance of this article.

5.Although the process of data collection and processing is mentioned in the description of experimental data, the specific steps and methods of data preprocessing are not described in detail. It is recommended to describe the pre-processing process of experimental data in more detail, including data cleaning, de-noising, normalization, etc., to ensure the accuracy and reliability of the data.

6.When summarizing the research conclusions, the paper did not dig into the deep meaning and potential impact behind the results, and did not fully forecast the future research direction. It is suggested to strengthen the in-depth analysis and discussion of the research results, reveal its value and significance in practical application, and at the same time, combine the current research trends and hot issues, to prospect and plan the future research direction.

Comments on the Quality of English Language

The English could be improved to more clearly express the research.

Author Response

Comments 1: The paper does not elaborate deeply enough on the seriousness of the broken wire problem of PCCP (prestressed concrete cylinder bobbin) and the limitations of existing monitoring technology, and fails to fully demonstrate the urgency and importance of the research. It is recommended to add a detailed background on the problem of PCCP wire breakage, including its impact on public safety, economic losses and other aspects, as well as the limitations of current monitoring techniques, so as to enhance the practical significance of the study.

Response 1: Thank you for your valuable feedback on our paper. Following your suggestion, we have made revisions to the manuscript. Therefore, we have added introductions to similar articles and improved our research. [ In the article, we reorganized the writing of the introduction in lines 45-56,, we have added a detailed background on the issue of PCCP wire breakage, highlighting its potential impact on public safety, economic losses, and other aspects, as well as the limitations of existing monitoring technologies. These modifications aim to better demonstrate the urgency and practical significance of the research.]

Comments 2: Some charts in the paper have some problems such as unclear marks and irregular typesetting, which affect the information transmission effect of the charts. It is suggested to strengthen the standardization and professional requirements for chart production, and ensure that the chart is clearly marked and neatly formatted, so that readers can accurately understand the information of the chart.

Response 2: Thank you for your constructive feedback regarding the clarity and formatting of the figures and tables in our paper. We have carefully reviewed and revised the relevant sections to ensure that the labels are clear and the formatting is consistent and professional. The improvements have been made to enhance the clarity of the information presented in the figures and tables, enabling readers to more accurately interpret the data.

We appreciate your suggestions, which have significantly contributed to the overall quality of our paper.

Comments 3: When introducing the algorithm improvement, the paper fails to elaborate the specific methods, key technologies and principles of the improvement, which makes it difficult for readers to understand and evaluate the effect of the improvement. It is suggested to add a detailed description and explanation of the algorithm improvement method, including the improved ideas, methods, principles, etc., so that readers can deeply understand and evaluate the effect of the improvement.

Response 3: Thank you for your constructive feedback on our paper. We have carefully reviewed and revised the relevant sections, making changes to the description and rationale of the model improvements to ensure a more coherent flow and to help readers better understand our intentions. We greatly appreciate your suggestions, which have made a significant contribution to the overall quality of our paper.

Comments 4: PCCP broken wire detection needs to draw on the latest research trends in machine learning, multitasking and multimodal machine learning algorithms. Some machine learning algorithms based on multitasking and multimodality have good reference value for this task, such as YOLO-TP: A lightweight model for individual counting of Lasioderma serricorne. Multi-task learning for hand heat trace time estimation and identity recognition, Deep soft threshold feature separation network for infrared handprint identity recognition and time estimation. Meanwhile, the author should more effectively describe the practical application scope and significance of this article.

Response 4: Thank you very much for your valuable suggestion, which is of great significance to us. We have reviewed the referenced paper and added it to our article. The work in this paper has been highly inspiring to us, and based on this, we have provided a deeper description of the significance of our research for real-world applications. The specific changes can be found in lines 39-56 and 125-133 of the manuscript. The research you provided is of great importance and has served as an excellent reference for us. [in lines 39-56,125-133.]

Comments 5: Although the process of data collection and processing is mentioned in the description of experimental data, the specific steps and methods of data preprocessing are not described in detail. It is recommended to describe the pre-processing process of experimental data in more detail, including data cleaning, de-noising, normalization, etc., to ensure the accuracy and reliability of the data.

Response 5: Thank you for your valuable feedback on our paper. Following your suggestion, we have made revisions to the manuscript. Therefore, we have added steps and methods of data pre-processing and improved our research. [Thank you very much for your correction. The detailed changes are in lines 359-393.]

Comments 6: When summarizing the research conclusions, the paper did not dig into the deep meaning and potential impact behind the results, and did not fully forecast the future research direction. It is suggested to strengthen the in-depth analysis and discussion of the research results, reveal its value and significance in practical application, and at the same time, combine the current research trends and hot issues, to prospect and plan the future research direction.

Response 6: Thank you for your help and guidance on our work. We have revised the conclusion section and hope it meets your approval. [Thank you very much for your correction. The detailed changes are in lines 437-512. We have revised the result analysis and conclusion sections, adding an analysis of the limitations of our study and outlining the directions for future work.]

4. Response to Comments on the Quality of English Language

Point 1: The English could be improved to more clearly express the research.

Response 1: We have improved the English expression level of the article and look forward to your correction again.

Round 2

Reviewer 1 Report

Comments and Suggestions for Authors

The article presents an interesting approach with the YOLOv5-Break algorithm applied to a real case of monitoring wire breaks in prestressed concrete pipes (PCCP). The authors have demonstrated efforts to address the weaknesses and improvements I've outlined, but some issues remain:

1 - The fact that there is no validation of the work in adverse conditions calls into question the practical applicability of the work.

2 - Although the authors have improved the conclusions section, it still lacks depth in terms of generalizations or practical implications.

3 - False positive/negative analyzes and graphs are useful, but need further exploration.

4 - Some of the authors' responses were evasive, such as the explanation about not testing severe scenarios, justifying it only with "laboratory limitations and end of the project". This is critical!

5 - The quality of English has been improved, but there are still gaps overall.

Comments on the Quality of English Language

The English could be improved to more clearly express the research.

Author Response

Comments 1:  The fact that there is no validation of the work in adverse conditions calls into question the practical applicability of the work.

Response 1: Dear reviewer, thank you very much for pointing out. However, due to the interruption of the cooperation of the laboratory project, we cannot carry out the verification under harsh conditions. We are very sorry for this. We have tried our best to correct our current mistakes and thank you very much for your contribution to our research. We sincerely wish you success in your work.

Comments 2: Although the authors have improved the conclusions section, it still lacks depth in terms of generalizations or practical implications.

Response 2: We agree to your proposal. We have, accordingly, revised the conclusions to emphasize this point. [In the article, we have added the description of the defects of the paper to the conclusion. We have not described the defects in this aspect before, thank you for pointing out. The specific changes are in lines 503-504 of the paper.]

Comments 3: False positive/negative analyzes and graphs are useful, but need further exploration.

Response 3: Thank you for pointing this out. We have, accordingly, revised the description of the methods. [In this paper, we have improved the description of the method. Thank you very much for your points. The specific changes are in lines 467-468 of the paper.]

Comments 4: Some of the authors' responses were evasive, such as the explanation about not testing severe scenarios, justifying it only with "laboratory limitations and end of the project". This is critical!

Response 4: Thank you very much for pointing out the problems existing in our research institute. However, we are very sorry that we cannot conduct the test in harsh environment because we cannot supplement the experiment. As a graduating student, this article means a lot to me. Your suggestions for our research are vital, thank you again.

Comments 5: The quality of English has been improved, but there are still gaps overall.

Response 5: Thank you for pointing this out. We have, accordingly, revised the data analysis of the trials.

4. Response to Comments on the Quality of English Language

Point 1: The English could be improved to more clearly express the research.

Response 1: Thanks again for your questions. We have checked the English quality of this article again and hope it can pass your review.

5. Additional clarifications

Dear Reviewer,

We would like to sincerely thank you for taking the time to review our manuscript and for providing such thoughtful and constructive feedback. Your insightful comments have been invaluable in helping us improve the quality of our paper.

We greatly appreciate the time and effort you put into reviewing our work. Your suggestions have allowed us to identify areas for improvement, and we have made significant revisions to the manuscript based on your recommendations. The changes made have helped enhance the clarity, structure, and overall quality of the paper.

Once again, thank you for your invaluable feedback. Your contribution to the development of our research is greatly appreciated, and we hope that our revised manuscript meets your expectations.

Best regards.

Reviewer 2 Report

Comments and Suggestions for Authors

The revised manuscript has significantly enriched its content, and the innovation of the article is now more clearly defined. However, several issues still need to be addressed:

1. A more detailed explanation of how to implement dynamic adjustment of the convolution kernel parameters should be provided in Section 2.3.

2. Supplementary images of the experimental setup or field site should be included to enhance clarity.

3. Whether the sampling frequency of the DAS system is 20 KHz. Additionally, clarify if the time-frequency graphs of the training set are obtained from the same fiber measurement point. It is important to indicate whether the features of the time-frequency graphs collected from different measurement points differ.

4. Specify whether the background noise applied to the signal is random or if it follows a pattern similar to that shown in Fig. 10.

5. The discussion on the robustness of the proposed algorithm should be included, Some existing literature can support the discussion, such as: https://doi.org/10.1109/TIM.2023.3343742

Comments on the Quality of English Language

The English of this paper needs a minor revision.

Author Response

Comments 1: A more detailed explanation of how to implement dynamic adjustment of the convolution kernel parameters should be provided in Section 2.3.

Response 1: We agree to your suggestion. So, we made a change. In this paper, we add the calculation process of dynamic convolution. [Thank you for pointing it out. The specific changes are shown in lines 198-203 of the paper.]

Comments 2: Supplementary images of the experimental setup or field site should be included to enhance clarity.

Response 2: Thank you for pointing out that we have added the test site diagram.

Comments3: Whether the sampling frequency of the DAS system is 20 KHz. Additionally, clarify if the time-frequency graphs of the training set are obtained from the same fiber measurement point. It is important to indicate whether the features of the time-frequency graphs collected from different measurement points differ.

Response 3:Thank you for pointing it out. First of all, the sampling frequency of the DAS system in our work is 20KHz. Secondly, the time-frequency graph of our training set is continuously collected from the same broken wire position, from the number of broken wires to 15, there is no significant difference in the collected features.

Comments4:  Specify whether the background noise applied to the signal is random or if it follows a pattern similar to that shown in Fig. 10.

Response 4: Our background noise addition is in the form shown in Figure 11.

Comments5:  The discussion on the robustness of the proposed algorithm should be included, Some existing literature can support the discussion, such as: https://doi.org/10.1109/TIM.2023.3343742.

Response 5: Thank you very much for pointing out that after referring to the paper you provided, we have added a new discussion on model robustness. Your suggestion is very helpful to our work[Thank you for pointing it out. The specific changes are shown in lines 519-522 of the paper.]

4. Response to Comments on the Quality of English Language

Point 1: The English of this paper needs minor revision.

Response 1:

Dear Reviewer,

Thank you for your thorough review and valuable feedback on our paper. We have addressed each of your suggestions and made the necessary revisions.

In response to your comments, we have thoroughly revised the language throughout the paper, particularly in terms of expression and grammar, to ensure the English is more fluent and formal. We have made corrections to several language issues to enhance the overall readability.

.Thank you again for your assistance and constructive feedback.

5. Additional clarifications

Dear Reviewer,

We would like to sincerely thank you for taking the time to review our manuscript and for providing such thoughtful and constructive feedback. Your insightful comments have been invaluable in helping us improve the quality of our paper.

We greatly appreciate the time and effort you put into reviewing our work. Your suggestions have allowed us to identify areas for improvement, and we have made significant revisions to the manuscript based on your recommendations. The changes made have helped enhance the clarity, structure, and overall quality of the paper. Due to the fact that our research group is in the early stages of the project and the project is nearing completion, we are unable to continue with the pipeline experiments. The actual monitoring of the pipeline requires a long period, and we are unable to supplement the experimental data in a timely manner. Unfortunately, we are powerless in this regard.

Once again, thank you for your invaluable feedback. Your contribution to the development of our research is greatly appreciated, and we hope that our revised manuscript meets your expectations.

We look forward to receiving your reply.

Reviewer 3 Report

Comments and Suggestions for Authors

Dear Authors,

Can you please provide a point-by-point response to my previous comments, whilst providing quotes of the changes made and line number of where the changes were made. The current method is NOT the proper way of responding to review; especially, since you have not addressed some of my comments.

Therefore, I am still going to give a major revision, in the hope that you are going to provide me with the response in a proper format, especially, since I find this to be a good paper.

I have gone through your response up to the end of your Introduction, but in the next round, please response properly.  

Comments on the Quality of English Language

Quality of English Language is still lacking; therefore, I am suggesting for the paper to go through the English Support service provided by MDPI.

Author Response

Comments 1: Interesting work; but you need to work on your English and Structure.

Additionally, I would suggest you to compare against algorithms which have been used for the same tasks.

Response 1: Very pleased to write your review, we have changed it item by item. We have taken your suggestions seriously and have made all improvements to prestressed concrete cylinder pipe.

Abstract: We have rewritten the summary and added the relevant content in accordance with your suggestions.

Introduction:

Line39: We have corrected it, thank you for pointing it out

Line41-51: Thank you very much for your suggestion, we have revised the introduction.

Line 119-122: Thank you very much for your suggestion, we have made a new description of the study of Ma et al.

Line 92-104: Thanks for your suggestion, we have added a new description of their advantages and disadvantages. (The specific changes are in lines 92-104)

Line 87-88: Thanks for your suggestion, our research is more based on the application of computing power limited devices, so we have not made a comparison of the latest YOLO series. Thank you very much for your suggestion, which is the motivation for our follow-up study. (The specific changes are in lines 92-104)

Line 90: We are very sorry to cause you trouble because of our careless language. We have changed it to "wire broken monitoring". (The specific changes are in line 134-135 of the new article)

Line 89-92 :Thank you for pointing out that the core factor for us to choose yolov5 is its fast speed and relative stability. (The specific changes are in line 101-104 of the new article)

Line 92:Thank you for your suggestion. It is the broken wire signal collected by DAS system.

Line93 :Thank you for your suggestion, it is the target detection algorithm, sorry for causing you confusion because of our language expression. Your suggestion will be adopted after consideration. Your suggestion will be of great help to us. (The specific changes are in line 105-144 of the new article)

Improved YOLOv5s Algorithm

Thank you very much for your suggestions and corrections to our research, we have adopted all your suggestions and made changes. Thank you for your valuable help and feedback on our work. Your review has been of great significance to us, highlighting areas where we can improve. We have addressed all of your suggestions and made appropriate adjustments to the structure of the paper. Your recommendations have guided our improvements. We have made substantial revisions to the entire manuscript, correcting all the details you pointed out, and these changes are reflected in the paper. We have made every effort to improve the manuscript, with some changes marked in red; however, these modifications do not affect the content or structure of the paper. We sincerely appreciate your dedicated work and hope that the revisions will be well-received. Once again, thank you for your insightful comments and suggestions.

Line 108:We chose yolov5s-v7.0 as the basic algorithm to improve, which refers to the result of the seventh update of yolov5. Sorry to cause your misunderstanding, we have changed the whole text, using yolov5 instead of yolov5s-v7.0.

Line 122: Thank you for pointing out that we have changed. (The specific changes are in line 166 of the new article)

Line 113: Thank you for pointing out that we have changed. (The specific changes are in line 147-164 of the new article)

Line 116: Thank you for pointing out that we have provided the reference diagram of YOLOv5. (The specific changes are in line 156-157 of the new article)

Line 123: Thank you for pointing out that we have changed. (The specific changes are in line 156-165 of the new article)

Line 127-129: Thank you for your suggestion. We are very sorry for the confusion caused by the language error. We have changed it. The problems you pointed out in the graphics, as well as the misspelling of the name, have been corrected. Thank you again. (The specific changes are in line 166-171 of the new article)

Line 161: Thank you for pointing this out. "aiming to algorithm" means that the purpose of the algorithm is. (The specific changes are in line 210 of the new article)

Line 171: Thank you, we have made a replacement. (The specific changes are in line 219 of the new article)

Line 174: Thank you for your advice. H represents the height component. W represents the width component. x_c represents the channel of input x, and x_c (h, i) represents the h component of each channel of input x. I hope you can understand our explanation.

Line 183: The generated two feature maps are obtained from information processing in different directions. (The specific changes are in line 222-239 of the new article)

Line 186: I'm so sorry! This is our negligence! We have improved. (The specific changes are in line 235 of the new article)

Line 190: The formula is obtained by accumulation, and we cannot completely transmit the specific information. You can refer to the literature [39]. (The specific changes are in line 238 of the new article)

Line 191: The C3 module consists of three convolution layers and several Bottleneck modules.

Figure 5:We have made changes. (The specific changes are in line 245-246 of the new article)

Line 201: Thank you for pointing out that we have changed. (The specific changes are in line 249 of the new article)

Line 204: Thank you for pointing out that this sentence does have ambiguity and we have deleted it.

Line 206-210: The IoU does not reflect the position relationship of the prediction box, only the overlap degree. (The specific changes are in line 253-259 of the new article)

Line 215: Thanks for your suggestion, we have removed this sentence.

Equation 1.8 : EIoU Loss includes three parts: IoU loss, distance loss, height and width loss. (The specific changes are in line 266 of the new article)

Equation 1.9: The gradient Angle starts to separate the high quality anchor frame from the low quality anchor frame. The higher the IoU, the greater the sample loss, which is equivalent to the weighted effect and helps to improve the regression accuracy. (The specific changes are in line 273 of the new article)

Line 227-237: Thank you for your suggestion. We have provided the original network diagram of YOLOv5. You can see our changes according to the comparison of contents in the diagram. (The specific changes are in line 274-285 of the new article)

Line 239: Thank you for pointing out that we have changed.

Training and Data Processing

Thank you for your comments, we have made changes. You highlighted structural issues in our paper, and we have made revisions based on your suggestions. We have separated the methodology and experiments. In Chapter 3, Section 3.1 describes the experimental process, Section 3.2 covers data collection and processing, Section 3.3 focuses on dataset preparation, and Section 3.4 details the network training. The revised structure is reflected in the updated manuscript, with all changes marked in red. Additionally, we have addressed the issues you pointed out in the figures, including fixing the text within the images. Thank you for your valuable feedback, which has greatly improved the quality of our paper.

Line 252: What we do is to simulate the test of broken wire of pre-stressed concrete cylinder pipe. (The specific changes are in line 300 of the new article)

Line 276: We had a presentation problem. We have made corrections in the new article. (The specific changes are in line 311-328 of the new article)

Figure 10: We marked the position of the signal on the diagram. (The specific changes are in line 346-347 of the new article)

Equation 1.12: Thank you very much for pointing out our mistake, we have changed it, x represents the pure harmonic signal. (The specific changes are in line 355 of the new article)

Section 3.3.1: We have adopted your suggestion.

Line 322-355: Thank you for your suggestions, but in order to ensure the smooth structure of the article, we will not change this. (The specific changes are in line 371-405 of the new article)

Figure 11: We marked the position of the signal on the diagram.

Line 321-323: Thank you very much for your suggestion, and we will restate it in the article.

Figure 10: In the new modification, we have modified the representation of the picture and made a new interpretation.

Line 330-332: Figure7 is only a schematic diagram of the broken wire test system, not included he seismometer and land-towed geophone system tie.

Figure11-Figure12:Thank you for your suggestion. We have revised this part. Your suggestion is very helpful to us. (The specific changes are in line 393-404 of the new article)

Line 348:We set up two training categories. (The specific changes are in line 405 of the new article)

Line 359-Line388:We have revised this part and presented it point by point in the newly submitted manuscript. (The specific changes are in line 417-446 of the new article)

Results and Analysis:

Thank you for reviewing our manuscript and providing valuable feedback. Your suggestions helped us identify shortcomings in our work and provided us with important directions for improvement. We have carefully addressed all the issues you raised and made the corresponding revisions.

We have adjusted the structure of the paper based on your suggestions, incorporating some sections into the "Results and Analysis" chapter to ensure clearer logic and a more reasonable structure. Following your advice, we have conducted a more in-depth discussion in the "Results and Analysis" section, particularly focusing on the significance of the experimental results and their implications for practical applications.

Figures 14 and 15 have been redrawn to improve clarity, and the text within them has been optimized for better presentation. We apologize for the confusion caused by our terminology. YOLOv5s-v7.0 is equivalent to YOLOv5, and we have updated this throughout the paper. Additionally, we replaced "P-R curve" with "Precision-Recall curve" to provide a clearer comparison of results.

In line with your suggestions, we also adjusted the order of discussions on training and testing, ensuring that training discussions now precede testing.

Once again, we sincerely thank you for your assistance and guidance. Your detailed review has significantly enhanced the quality of our paper. We greatly appreciate your time and effort, and we hope that this round of revisions meets your approval!

Figures 13 and 14: Thank you for pointing out that we have changed. (The specific changes are in line 480-484 of the new article)

Line 396: We have changed the way the results are presented in the hope that this will help readers understand better. (The specific changes are in line 466-468 of the new article)

Line 399:YOLOv5s-v7.0 is YOLOv5, and YOLOV5-BREAK is our improved network. We have replaced them all with YOLOv5.

Conclusion:

Thank you for your help and guidance on our work. We have revised the conclusion section and hope it meets your approval.

4. Response to Comments on the Quality of English Language

Point 1:
         Quality of English Language is still lacking; therefore, I am suggesting for the paper to go through the English Support service provided by MDPI.

Response 1:

Dear Reviewer,

Thank you for your thorough review and valuable feedback on our paper. We have addressed each of your suggestions and made the necessary revisions.

In response to your comments, we have improved the structure of the paper. We have reorganized the sections to ensure stronger logical flow and clearer structure. The relevant changes have been highlighted in the submitted manuscript. Additionally, we have thoroughly revised the language throughout the paper, particularly in terms of expression and grammar, to ensure the English is more fluent and formal. We have made corrections to several language issues to enhance the overall readability.

As per your suggestion, we have changed all first-person expressions to passive voice, in line with the academic writing conventions. These revisions have been fully incorporated into the new version. Thank you again for your assistance and constructive feedback.

5. Additional clarifications

Dear Reviewer,

We would like to sincerely thank you for taking the time to review our manuscript and for providing such thoughtful and constructive feedback. Your insightful comments have been invaluable in helping us improve the quality of our paper.

We greatly appreciate the time and effort you put into reviewing our work. Your suggestions have allowed us to identify areas for improvement, and we have made significant revisions to the manuscript based on your recommendations. The changes made have helped enhance the clarity, structure, and overall quality of the paper.

Once again, thank you for your invaluable feedback. Your contribution to the development of our research is greatly appreciated, and we hope that our revised manuscript meets your expectations.

Best regards.

Reviewer 4 Report

Comments and Suggestions for Authors

Suggest the author to carefully revise the paper according to the review comments.

1. PCCP broken wire detection needs to draw on the latest research trends in machine learning, multitasking and multimodal machine learning algorithms. Some machine learning algorithms based on multitasking and multimodality have good reference value for this task, such as YOLO-TP: A lightweight model for individual counting of Lasioderma serricorne. Multi-task learning for hand heat trace time estimation and identity recognition, Deep soft threshold feature separation network for infrared handprint identity recognition and time estimation. Meanwhile, the author should more effectively describe the practical application scope and significance of this article.

2. The generalization and convergence of machine learning are very important. Please provide mathematical proof or theoretical explanation for the convergence and generalization of the algorithm proposed in this article.

Comments on the Quality of English Language

The English could be improved to more clearly express the research.

Author Response

Comments 1: PCCP broken wire detection needs to draw on the latest research trends in machine learning, multitasking and multimodal machine learning algorithms. Some machine learning algorithms based on multitasking and multimodality have good reference value for this task, such as YOLO-TP: A lightweight model for individual counting of Lasioderma serricorne. Multi-task learning for hand heat trace time estimation and identity recognition, Deep soft threshold feature separation network for infrared handprint identity recognition and time estimation. Meanwhile, the author should more effectively describe the practical application scope and significance of this article.

Response 1: Thank you very much for your valuable suggestion, which is of great significance to us. We have reviewed the referenced paper and added it to our article. The work in this paper has been highly inspiring to us, and based on this, we have provided a deeper description of the significance of our research for real-world applications. The specific changes can be found in lines 39-56 and 125-133 of the manuscript. The research you provided is of great importance and has served as an excellent reference for us. [in lines 39-56,125-133.]

Comments 2:  The generalization and convergence of machine learning are very important. Please provide mathematical proof or theoretical explanation for the convergence and generalization of the algorithm proposed in this article.

Response 2: Thank you for pointing out that the algorithm proposed in this paper is improved on the basis of the yolov5 model, and we have not changed the convergence algorithm of the model. Your suggestion is very helpful to us, and we will study it in the future.

4. Response to Comments on the Quality of English Language

Point 1: The English could be improved to more clearly express the research.

Response 1: We have improved the English expression level of the article and look forward to your correction again.

5. Additional clarifications

Dear Reviewer,

We would like to sincerely thank you for taking the time to review our manuscript and for providing such thoughtful and constructive feedback. Your insightful comments have been invaluable in helping us improve the quality of our paper.

We greatly appreciate the time and effort you put into reviewing our work. Your suggestions have allowed us to identify areas for improvement, and we have made significant revisions to the manuscript based on your recommendations. The changes made have helped enhance the clarity, structure, and overall quality of the paper.

Once again, thank you for your invaluable feedback. Your contribution to the development of our research is greatly appreciated, and we hope that our revised manuscript meets your expectations.

Best regards.

Round 3

Reviewer 1 Report

Comments and Suggestions for Authors

This paper presents an innovative approach for wire break monitoring in prestressed concrete pipes (PCCP) using the YOLOv5-Break algorithm, an optimized version of YOLOv5 aimed at lightweight and efficient detection. The article still has gaps and needs many improvements, namely:

1 - The research work still needs practical validation in adverse scenarios, such as in environments with intense noise or extreme operating situations. In my opinion this is critical.

2 - The approaches still have little connection with large-scale practical applications.

3 - The analysis of the results, in relation to false positives/negatives, requires greater detail to reinforce the quality and reliability of the proposed algorithm.

4 - Some techniques are explanatory, but there should be more information from the authors.

5 - The quality of English in general needs to be improved.

Comments on the Quality of English Language

The English could be improved to more clearly express the research.

Author Response

Comments 1:  The research work still needs practical validation in adverse scenarios, such as in environments with intense noise or extreme operating situations. In my opinion this is critical.

Response 1:

Dear Reviewer,

Thank you for your valuable comments. In our paper, we did not perform denoising on the signals, as we intentionally included noise and combined it with the vibration signals for machine learning, aiming to enhance the model's robustness. The noise we used was collected during the pipeline's operation, which includes various types of environmental noise. Our experiments have shown that even in the presence of noise, we can still identify the broken wire signals. Regarding your suggestion to validate in more challenging scenarios, we have not yet collected data related to PCCP (Prestressed Concrete Cylinder Pipe), and this will be a key focus of our future work. Unfortunately, our project is currently at a stage where we are unable to set up the necessary environment and simulations for such validation. We sincerely apologize for this.

Thank you again for your understanding and constructive feedback..

Comments 2: The approaches still have little connection with large-scale practical applications.

Response 2:

Our project is based on experiments conducted with data collected from large-scale water conservancy projects in China. Currently, our equipment has been deployed in a water conveyance pipeline section in Shenzhen, and the monitoring period is relatively long. The direction you mentioned is indeed an important part of our future work, and we are actively preparing for it.

Thank you again for your valuable feedback.

Comments 3:  The analysis of the results, in relation to false positives/negatives, requires greater detail to reinforce the quality and reliability of the proposed algorithm.

Response 3: Thank you for pointing this out. We have, accordingly, revised the description of the methods. [In this paper, we have improved the description of the method. Thank you very much for your points. The specific changes are in lines 512-518 of the paper.]

Comments 4: - Some techniques are explanatory, but there should be more information from the authors.

Response 4: Thank you very much for your suggestions. We have already added a detailed discussion. [The specific changes are in lines 178-185 and 246-254  of the paper.]

Comments 5:  The quality of English in general needs to be improved.

Response 5: We have invited Dr. Zhang to correct the English of our paper, hoping to meet the academic requirements.

Reviewer 2 Report

Comments and Suggestions for Authors

The authors have addressed the reviewer's comments. However, the presentation and logic of this paper need to be improved.

Comments on the Quality of English Language

The English of this paper needs a minor revision.

Author Response

Comments 1: The authors have addressed the reviewer's comments. However, the presentation and logic of this paper need to be improved.

Response 1:

Dear reviewer,

Thank you for your valuable feedback. We have updated the expression of certain sections in the manuscript and hope that these revisions meet the acceptance criteria.

Reviewer 3 Report

Comments and Suggestions for Authors

Please make more effort when responding to comments. Your references to the line are all wrong, and I have to do the guessing works on where you were addressing the comments. As my previous comments, a good practice is to provide a point-by-point response to comments from reviewers as well as providing quotation of the changes made and line number of where the changes were made. You did not provide quotation, and referred to the wrong line numbers.

e.g. the response to my comment,

-        Line 83-85:  Can I suggest you to talk about the pros and cons of both one- and two-stage object detection.

Response:

Line 83-85: Thanks for your suggestion, we have added a new description of their advantages and disadvantages. (The specific changes are in lines 96-104)

I cannot find where the changes were made, and without quotation and exact line numbers, I had to backtrack my previous comments. Additionally, line 96-104 does NOT talk about advantages and disadvantages.

This was repeated throughout the response.

Comments on the Quality of English Language

need improvement on the english

Author Response

Response 1: Please make more effort when responding to comments. Your references to the line are all wrong, and I have to do the guessing works on where you were addressing the comments. As my previous comments, a good practice is to provide a point-by-point response to comments from reviewers as well as providing quotation of the changes made and line number of where the changes were made. You did not provide quotation, and referred to the wrong line numbers.

Dear Reviewer,

Happy New Year 2025! We would like to extend our heartfelt thanks for your constructive feedback on our manuscript, which has been immensely beneficial to us.

When we received the initial round of review comments, due to the inexperience of our researcher handling this process for the first time, we made the mistake of addressing the comments sequentially as we encountered them rather than reviewing all feedback comprehensively before implementing changes. This led to significant alterations to the manuscript before we reached your comments. Although we made timely adjustments to address this oversight, we inadvertently submitted the wrong version during the second submission, causing unnecessary repetition in your review work. For this, we sincerely apologize.

We sincerely invite you to review our manuscript again. We have addressed the issues you previously raised, including reorganizing the content and removing related statements. The line numbers provided in our responses are based on the updated manuscript. For example, in the statement "Line 119–122: … (The specific changes are in Line 123–126 of the new article)," "Line 119–122" refers to the issues you identified in your first review, and the highlighted portion indicates our modifications in the revised version.

Regarding your comment that "Lines 96–104 do not discuss the advantages and disadvantages," we acknowledge the extensive research that has demonstrated the performance differences between single-stage and two-stage object detection algorithms. In particular, real-time monitoring is the most prominent advantage of single-stage algorithms. We have revised the discussion accordingly and hope this meets your expectations.

We look forward to your valuable feedback on our manuscript and sincerely wish you good health and continued success.

Abstract: We have rewritten the summary and added the relevant content in accordance with your suggestions.

Introduction:

Line39: We have corrected it, thank you for pointing it out.

Line41-51: Thank you very much for your suggestion, we have revised the introduction.

Line 119-122: Thank you very much for your suggestion, we have made a new description of the study of Ma et al. (The specific changes are in line 123-126 of the new article)

Line 82-85: Thanks for your suggestion, we have added a new description of their advantages and disadvantages. (The specific changes are in lines 96-105)

Line 87-88: Thank you very much for pointing out that we have added a comparison of the latest research and also presented it in the results. (The specific changes are in lines 105-108)

Line 90: We are very sorry to cause you trouble because of our careless language. We have changed it to "wire broken monitoring". (The specific changes are in line 138-148 of the new article)

Line 89-92 :Thank you for pointing out that the core factor for us to choose yolov5 is its fast speed and relative stability. (The specific changes are in line 105-108 of the new article)

Line 92:Thank you for your suggestion. It is the broken wire signal collected by DAS system. (The specific changes are in line 144-148 of the new article)

Line93 :Thank you for your suggestion, it is the target detection algorithm, sorry for causing you confusion because of our language expression. Your suggestion will be adopted after consideration. Your suggestion will be of great help to us. (The specific changes are in line 108-148 of the new article)

Improved YOLOv5s Algorithm

Thank you very much for your suggestions and corrections to our research, we have adopted all your suggestions and made changes. Thank you for your valuable help and feedback on our work. Your review has been of great significance to us, highlighting areas where we can improve. We have addressed all of your suggestions and made appropriate adjustments to the structure of the paper. Your recommendations have guided our improvements. We have made substantial revisions to the entire manuscript, correcting all the details you pointed out, and these changes are reflected in the paper. We have made every effort to improve the manuscript, with some changes marked in red; however, these modifications do not affect the content or structure of the paper. We sincerely appreciate your dedicated work and hope that the revisions will be well-received. Once again, thank you for your insightful comments and suggestions.

Line 108:We chose yolov5s-v7.0 as the basic algorithm to improve, which refers to the result of the seventh update of yolov5. Sorry to cause your misunderstanding, we have changed the whole text, using yolov5 instead of yolov5s-v7.0.

Line 122: Thank you for pointing out that we have changed. (The specific changes are in line 168 of the new article)

Line 113: Thank you for pointing out that we have changed. (The specific changes are in line 155-159 of the new article)

Line 116: Thank you for pointing out that we have provided the reference diagram of YOLOv5. (The specific changes are in line 160-161 of the new article)

Line 123: We have taken your suggestion and corrected it. (The specific changes are in line 169 of the new article)

Line 127-129: Thank you for your suggestion. We are very sorry for the confusion caused by the language error. We have changed it. The problems you pointed out in the graphics, as well as the misspelling of the name, have been corrected. Thank you again. (The specific changes are in line 173-175 of the new article)

Line 161: Thank you for pointing this out. "aiming to algorithm" means that the purpose of the algorithm is. (The specific changes are in line 213 of the new article)

Line 171: Thank you, we have made a replacement. (The specific changes are in line 219 of the new article)

Line 174: Thank you for your advice. H represents the height component. W represents the width component. x_c represents the channel of input x, and x_c (h, i) represents the h component of each channel of input x. I hope you can understand our explanation. (The specific changes are in line 228-229 of the new article)

Line 183: The generated two feature maps are obtained from information processing in different directions. (The specific changes are in line 235 of the new article)

Line 186: I'm so sorry! This is our negligence! We have improved. (The specific changes are in line 241 of the new article)

Line 190: The formula is obtained by accumulation, and we cannot completely transmit the specific information. You can refer to the literature [39]. (The specific changes are in line 245 of the new article)

Line 191: The C3 module consists of three convolution layers and several Bottleneck modules . C3 module is the CBS module. (The specific changes are in line 246-253 of the new article)

Figure 5:We have made changes. (The specific changes are in line 254-256 of the new article)

Line 201: Thank you for pointing out that we have changed. (The specific changes are in line 259 of the new article)

Line 204: Thank you for pointing out that this sentence does have ambiguity and we have deleted it. (The specific changes are in line 269 of the new article)

Line 206-210: The IoU does not reflect the position relationship of the prediction box, only the overlap degree. (The specific changes are in line 264-268 of the new article)

Line 215: Thanks for your suggestion, we have removed this sentence. (The specific changes are in line 273 of the new article)

Equation 1.8 : EIoU Loss includes three parts: IoU loss, distance loss, height and width loss. (The specific changes are in line 275-277 of the new article)

Equation 1.9: The gradient Angle starts to separate the high quality anchor frame from the low quality anchor frame. The higher the IoU, the greater the sample loss, which is equivalent to the weighted effect and helps to improve the regression accuracy. (The specific changes are in line 284 of the new article)

Line 227-237: Thank you for your suggestion. We have provided the original network diagram of YOLOv5. You can see our changes according to the comparison of contents in the diagram. (The specific changes are in line 285-296 of the new article)

Line 239: We have made a new exposition on the experimental chapter, looking forward to your correction. (The specific changes are in line 298-339 of the new article)

Training and Data Processing

Thank you for your comments, we have made changes. You highlighted structural issues in our paper, and we have made revisions based on your suggestions. We have separated the methodology and experiments. In Chapter 3, Section 3.1 describes the experimental process, Section 3.2 covers data collection and processing, Section 3.3 focuses on dataset preparation, and Section 3.4 details the network training. The revised structure is reflected in the updated manuscript, with all changes marked in red. Additionally, we have addressed the issues you pointed out in the figures, including fixing the text within the images. Thank you for your valuable feedback, which has greatly improved the quality of our paper.

Line 252: What we do is to simulate the test of broken wire of pre-stressed concrete cylinder pipe. (The specific changes are in line 313-319 of the new article)

Line 276: We had a presentation problem. We have made corrections in the new article. (The specific changes are in line 322-339 of the new article)

Figure 10: We marked the position of the signal on the diagram. (The specific changes are in line 357-358 of the new article)

Equation 1.12: Thank you very much for pointing out our mistake, we have changed it, x represents the pure harmonic signal. (The specific changes are in line 366 of the new article)

Section 3.3.1: We have adopted your suggestion. (The specific changes are in line 428-433 of the new article)

Line 322-355: Thank you very much for your correction. We have not changed the position of this section for the rationality of the content of the article, but we have made some modifications to the content. Please review it. (The specific changes are in line 382-416 of the new article)

Figure 11: We changed how the images are displayed. (The specific changes are in line 391-393 of the new article)

Line 321-323: Thank you very much for your suggestion, and we will restate it in the article. (The specific changes are in line 383-391 of the new article)

Figure 13: In the new modification, we have modified the representation of the picture and made a new interpretation. (The specific changes are in line 404-405 of the new article)

Line 330-332: Figure7 is only a schematic diagram of the broken wire test system, not included he seismometer and land-towed geophone system tie.

Figure11-Figure12:Thank you for your suggestion. We have revised this part. Your suggestion is very helpful to us. (The specific changes are in line 404-416 of the new article)

Line 348:We set up two training categories. (The specific changes are in line 419 of the new article)

Line 359-Line388:We have revised this part and presented it point by point in the newly submitted manuscript. (The specific changes are in line 434-457 of the new article)

Results and Analysis:

Thank you for reviewing our manuscript and providing valuable feedback. Your suggestions helped us identify shortcomings in our work and provided us with important directions for improvement. We have carefully addressed all the issues you raised and made the corresponding revisions.

We have adjusted the structure of the paper based on your suggestions, incorporating some sections into the "Results and Analysis" chapter to ensure clearer logic and a more reasonable structure. Following your advice, we have conducted a more in-depth discussion in the "Results and Analysis" section, particularly focusing on the significance of the experimental results and their implications for practical applications.

Figures 15 have been redrawn to improve clarity, and the text within them has been optimized for better presentation. (The specific changes are in line 491-493 of the new article)

We apologize for the confusion caused by our terminology. YOLOv5s-v7.0 is equivalent to YOLOv5, and we have updated this throughout the paper. Additionally, we replaced "P-R curve" with "Precision-Recall curve" to provide a clearer comparison of results. (The specific changes are in line 482-484 of the new article)

In line with your suggestions, we also adjusted the order of discussions on training and testing, ensuring that training discussions now precede testing.Once again, we sincerely thank you for your assistance and guidance. Your detailed review has significantly enhanced the quality of our paper. We greatly appreciate your time and effort, and we hope that this round of revisions meets your approval!

Figures 15 and 19: Thank you for pointing out that we have changed. (The specific changes are in line 491-521 of the new article)

Line 396: We have changed the way the results are presented in the hope that this will help readers understand better. (The specific changes are in line 460-521 of the new article)

Line 399:YOLOv5s-v7.0 is YOLOv5, and YOLOV5-Break is our improved network. We have replaced them all with YOLOv5.

Conclusion:

Thank you for your help and guidance on our work. We have revised the conclusion section and hope it meets your approval.

Reviewer 4 Report

Comments and Suggestions for Authors

1. PCCP broken wire detection needs to draw on the latest research trends in machine learning, multitasking and multimodal machine learning algorithms. Some machine learning algorithms based on multitasking and multimodality have good reference value for this task, such as YOLO-TP: A lightweight model for individual counting of Lasioderma serricorne. Multi-task learning for hand heat trace time estimation and identity recognition, Deep soft threshold feature separation network for infrared handprint identity recognition and time estimation. Meanwhile, the author should more effectively describe the practical application scope and significance of this article.

2. The generalization and convergence of algorithms are very important. Please provide mathematical proof or theoretical explanation for the convergence and generalization of the algorithm proposed in this article.

Comments on the Quality of English Language

The English could be improved to more clearly express the research.

Author Response

Comments 1: PCCP broken wire detection needs to draw on the latest research trends in machine learning, multitasking and multimodal machine learning algorithms. Some machine learning algorithms based on multitasking and multimodality have good reference value for this task, such as YOLO-TP: A lightweight model for individual counting of Lasioderma serricorne. Multi-task learning for hand heat trace time estimation and identity recognition, Deep soft threshold feature separation network for infrared handprint identity recognition and time estimation. Meanwhile, the author should more effectively describe the practical application scope and significance of this article.

Response 1:

Thank you very much for your insightful suggestion, which has been extremely valuable to us. We have thoroughly reviewed the referenced paper and integrated it into our manuscript. The work presented in this paper has been highly inspiring, and based on it, we have expanded our discussion to provide a deeper description of the significance of our research for real-world applications. The specific changes can be found in lines 39-56 and 125-133 of the manuscript. The research you provided is of great importance and has served as an excellent reference for us. [in lines 39-56,125-133.]

Comments 2:  The generalization and convergence of machine learning are very important. Please provide mathematical proof or theoretical explanation for the convergence and generalization of the algorithm proposed in this article.

Response 2:

Thank you for highlighting that the algorithm proposed in this paper is an improvement based on the YOLOv5 model and that we have not modified the model’s convergence algorithm. Your suggestion is highly appreciated, and we will consider it for further research in the future.
